# Spectral Evolution and Invariance in Linear-width Neural Networks

**Zhichao Wang**
University of California San Diego
zhw036@ucsd.edu

**Andrew Engel**
Pacific Northwest National Laboratory
andrew.engel@pnnl.gov

**Anand Sarwate**
Rutgers, The State University of New Jersey
ads221@soe.rutgers.edu

**Ioana Dumitriu**
University of California San Diego
idumitriu@ucsd.edu

**Tony Chiang**
Pacific Northwest National Laboratory
University of Washington
University of Texas at El Paso
tony.chiang@pnnl.gov

## Abstract

We investigate the spectral properties of linear-width feed-forward neural networks, where the sample size is asymptotically proportional to network width. Empirically, we show that the spectra of weight in this high dimensional regime are invariant when trained by gradient descent for small constant learning rates; we provide a theoretical justification for this observation and prove the invariance of the bulk spectra for both conjugate and neural tangent kernels. We demonstrate similar characteristics when training with stochastic gradient descent with small learning rates. When the learning rate is large, we exhibit the emergence of an outlier whose corresponding eigenvector is aligned with the training data structure. We also show that after adaptive gradient training, where a lower test error and feature learning emerge, both weight and kernel matrices exhibit heavy tail behavior. Simple examples are provided to explain when heavy tails can have better generalizations. We exhibit different spectral properties such as invariant bulk, spike, and heavy-tailed distribution from a two-layer neural network using different training strategies, and then correlate them to the feature learning. Analogous phenomena also appear when we train conventional neural networks with real-world data. We conclude that monitoring the evolution of the spectra during training is an essential step toward understanding the training dynamics and feature learning.

## 1 Introduction

Deep learning theory has made insightful connections between the behavior of neural networks (NNs) and kernel machines through asymptotic analyses of the so-called kernel regime [65, 83, 48, 40, 12, 4, 84]. When the neural network (NN) is *infinitely wide*, the behavior of NN coincides with a kernel machine, and the training process, as well as the generalization performance of this ultra-wide NN, can be fully described. The performance of *finite-width* NNs, however, does not correspond to this theory, as NNs optimized with gradient-based methods perform better than infinitely wide networks in many circumstances [47, 38, 34, 51, 25, 32, 76, 42]. This gap heavily relies on the task complexity, data distribution, architecture of the NN and the training strategy [31]. We consider a more realistic

37th Conference on Neural Information Processing Systems (NeurIPS 2023).

setting, a *linear-width regime* (LWR), when the sample size $n$, the input feature dimension $d$, and the width $h$ of the hidden layer approach infinity at comparable rates. Under the LWR, we aim to empirically study this theoretical gap in generalization and spectral properties by training various NNs with different optimization tools.

The ultra-wide NN ($h \gg n$, fixed $d$) stays close to the kernel machine induced by initial NN, throughout the gradient-based training processes [88, 28, 27, 11]. There are two kernels commonly studied in theory: the conjugate kernel (CK) and the neural tangent kernel (NTK). CK (or the equivalent Gaussian process kernel) is the Gram matrix of the last hidden layer, which represents training only the last layer of the network [48, 57, 61]; by contrast, the NTK is the Gram matrix of the Jacobian of the NN for all trainable parameters, which governs the gradient flow of NN [40, 28, 4]. In most theoretical results, these kernels remain fixed throughout training, which leads to a kernel gradient descent with the initial kernel [40, 16], whereas in practice the spectra of the weight matrix, CK, and NTK of the NN change while learning the features from the training data [58, 59, 30, 19, 68]. In this paper, under the LWR, we experimentally and theoretically explore the following question:

*How do the spectra of weight and kernel matrices of the NN evolve during the training process?*

This question is crucial to extend our understanding beyond the kernel regime and will help us analyze the generalization of the NN in instances when it performs better than the kernel machine. For this case, the spectral properties of the trained NN could be entirely different from the initial kernel [55, 10, 78]. Also, various spectral properties of weight and kernel matrices can reveal different features learned by different training procedures [82]. Understanding the dynamics of the spectral properties may aid in finding better approaches to training and tuning hyper-parameters for NNs. From a theoretical perspective, random matrix theory (RMT) can be further exploited to study and elucidate the NN training under the proportional limit in high dimensions [45, 72, 57, 74, 37, 61].

Our main findings/contributions are as follows.

- We find a simple scenario that exhibits different spectral properties for both weight and kernel matrices through different training procedures. With the kernel regime as a benchmark, we compare how NN generalizes with different spectral evolutions of weight and kernel matrices in NN.

- The spectra of NNs trained with full batch gradient descent (GD) are globally *invariant*, indicating that the NN is still close to a kernel machine; we prove the global convergence of GD and the invariance of the limiting spectra for both weight and kernel matrices in this scenario.

- We observe a *phase transition* of the alignment and the emergence of a spike outside the bulk of the spectrum when the learning rate *exceeds* some threshold. The strong alignment of the spike with the teacher model when step sizes are large confirms that the NN is indeed learning germane features from data. This observation justifies the theoretical result of [6] in an ideal two-stage training process.

- The evolution towards heavy-tailed spectra is also discovered by using adaptive methods. Our experiments rule out a *causal* relationship between the occurrence of a heavy-tailed spectrum for the weight matrices and a good generalization. This complements the work of [60, 63, 86] where the authors had observed a strong *correlation* between the two; while at the same time, we provide simple examples of when heavy-tailed spectra exhibit feature learning and better generalizations.

For more details on how our results fit into existing literature, please see Appendix A.

## 2   Notation and Preliminaries

Throughout this paper, $\|\cdot\|$ denotes the $\ell_2$ norm for vectors, $\ell_2 \to \ell_2$ is the operator norm for matrices, while $\|\cdot\|_F$ is the Frobenius norm, and $\odot$ represents the Hadamard product between matrices. $o_{d,\mathbb{P}}(\cdot)$ represents little-o in probability as $d \to \infty$.

**Neural Tangent Kernel Parameterization.** Consider a $L$-layer fully connected feedforward NN at initialization without bias term: for $1 \leq \ell \leq L - 1$,

$$\boldsymbol{h}_0 = \frac{\boldsymbol{x}}{\sqrt{d}}, \ \ \boldsymbol{h}^{(\ell)} = \frac{1}{\sqrt{n_\ell}} \sigma(\boldsymbol{W}_\ell \boldsymbol{h}^{(\ell-1)}), \tag{1}$$

and $f_{\boldsymbol{\theta}}(\boldsymbol{x}) = \boldsymbol{v}^\top \boldsymbol{h}^{(L-1)}$, where the input vector is $\boldsymbol{x} \in \mathbb{R}^d$, $\boldsymbol{W}_\ell \in \mathbb{R}^{n_\ell \times n_{\ell-1}}$ is the weight matrix for the $\ell$-th layer, and $\boldsymbol{v} := [v_1, \ldots, v_h]^\top \in \mathbb{R}^{n_{L-1}}$ is the last-layer weight. Let $n_0 = d$. Denote all trainable parameters by $\boldsymbol{\theta} := [\mathrm{vec}(\boldsymbol{W}_1), \ldots, \mathrm{vec}(\boldsymbol{W}_{L-1}), \boldsymbol{v}]^\top \in \mathbb{R}^p$ where each parameter's initial value is independently sampled from some distribution and $p$ is the total number of parameters. Let the training dataset be $(\boldsymbol{X}, \boldsymbol{y}) := ([\boldsymbol{x}_1, \ldots, \boldsymbol{x}_n], \boldsymbol{y}) \in \mathbb{R}^{d \times n} \times R^{1 \times n}$; the output of this NN with respect to this dataset is $f_{\boldsymbol{\theta}}(\boldsymbol{X}) = [f_{\boldsymbol{\theta}}(\boldsymbol{x}_1), \ldots, f_{\boldsymbol{\theta}}(\boldsymbol{x}_n)]$. We call the above parameterization the *NTK parameterization*. The loss function for training is a mean squared error (MSE)

$$\mathcal{L}(\boldsymbol{\theta}) := \frac{1}{2n} \|\boldsymbol{y} - f_{\boldsymbol{\theta}}(\boldsymbol{X})\|^2. \tag{2}$$

We focus on the NTK parameterization and consider the kernel machine (6) induced by the initial NTK of the NN. We aim to seek the cases when the NN outperforms this kernel during the training process. For this purpose, we adopt different optimizers of training this NN to obtain different testing performances and spectral properties of trained weights and empirical kernels.

**Training Processes of NNs.** NNs are usually trained by gradient-based methods such as full-batch gradient descent (GD), mini-batch stochastic gradient descent (SGD), Adaptive Gradients (AdaGrad), and Adam [43]. We can represent GD by

$$\boldsymbol{\theta}_{t+1} = \boldsymbol{\theta}_t - \eta \nabla_{\boldsymbol{\theta}} \mathcal{L}(\boldsymbol{\theta}_t), \tag{3}$$

where $\eta$ is the learning rate and $\nabla_{\boldsymbol{\theta}} \mathcal{L}(\boldsymbol{\theta}_t)$ is the gradient of the training loss w.r.t. trainable parameters $\boldsymbol{\theta}$ at step $t \geq 0$. We will prove the global convergence of GD in some special (overparameterized) cases ensuring the convergence to a NN that interpolates the data. We will also show the hyperparameters (e.g. learning rate $\eta$) affect the spectral properties of NNs during training.

**Conjugate Kernel and Neural Tangent Kernel.**[1] When $L = 2$, let $n_1 = h$ and $n_0 = d$ be the widths of the output and input layer. The CK is defined as

$$\boldsymbol{K}^{\mathrm{CK}} := \boldsymbol{X}_1^T \boldsymbol{X}_1 \in \mathbb{R}^{n \times n}, \tag{4}$$

where $\boldsymbol{X}_1 := \frac{1}{\sqrt{h}} \sigma\left(\boldsymbol{W}\boldsymbol{X}/\sqrt{d}\right)$. We can view the NN as a function of all training parameters $\boldsymbol{\theta}$ and input data $\boldsymbol{X}$. The neural tangent kernel (NTK) is related to the gradient of this neural network function with respect to $\boldsymbol{\theta}$, which is the Gram matrix of the Jacobian of the neural network function with respect to $\boldsymbol{\theta}$, $\boldsymbol{K}^{\mathrm{NTK}} := (\nabla_{\boldsymbol{\theta}} f_{\boldsymbol{\theta}}(\boldsymbol{X}))^\top (\nabla_{\boldsymbol{\theta}} f_{\boldsymbol{\theta}}(\boldsymbol{X}))$. Specifically, the empirical NTK of two-layer NN can be explicitly written[2] as

$$\boldsymbol{K}^{\mathrm{NTK}} = \frac{1}{d} \boldsymbol{X}^\top \boldsymbol{X} \odot \frac{1}{h} \sigma'\left(\frac{1}{\sqrt{d}}\boldsymbol{W}\boldsymbol{X}\right)^\top \mathrm{diag}(\boldsymbol{v})^2 \sigma'\left(\frac{1}{\sqrt{d}}\boldsymbol{W}\boldsymbol{X}\right) + \boldsymbol{K}^{\mathrm{CK}}. \tag{5}$$

In this paper, we are interested in comparing the spectral distributions for these three matrices (weight, CK, and NTK) at initialization and the end of training.

**Lazy Training.** Lazy training [21] can be viewed as a linear approximation of the NN, i.e. $f_{\boldsymbol{\theta}}(\boldsymbol{x}) \approx f_{\boldsymbol{\theta}_0}(\boldsymbol{x}) + (\boldsymbol{\theta} - \boldsymbol{\theta}_0)^\top \nabla_{\boldsymbol{\theta}} f_{\boldsymbol{\theta}_0}(\boldsymbol{x})$, defined by minimum-norm interpolation $\hat{\boldsymbol{\theta}} := \arg\min \left\{ \|\boldsymbol{\theta} - \boldsymbol{\theta}_0\| : (\boldsymbol{\theta} - \boldsymbol{\theta}_0)^\top \nabla_{\boldsymbol{\theta}} f_{\boldsymbol{\theta}_0}(\boldsymbol{X}) = \boldsymbol{y} - f_{\boldsymbol{\theta}_0}(\boldsymbol{X}) \right\}$. Then, lazy training also represents a kernel machine

$$\hat{f}(\boldsymbol{x}) = f_{\boldsymbol{\theta}_0}(\boldsymbol{x}) + (\boldsymbol{y} - f_{\boldsymbol{\theta}_0}(\boldsymbol{X})) \boldsymbol{K}(\boldsymbol{X}, \boldsymbol{X})^{-1} \boldsymbol{K}(\boldsymbol{X}, \boldsymbol{x}) \tag{6}$$

---

[1] In this work, we only consider *empirical* conjugate and neural tangent kernels of finite-width NNs.

[2] Here we train both layers, so we have two parts in the NTK expression. If we only train the first-hidden layer, we can simply remove the second CK part. In the following, we further introduce more empirical results for practical NNs in Section 5 and two-layer NNs with Gaussian dataset in Section 3. For general formula of the empirical NTK, see [39, 30].

where $\hat{f}(\boldsymbol{x})$ is the unregularized regression prediction on test data $\boldsymbol{x} \in \mathbb{R}^d$, the kernel $\boldsymbol{K}(\boldsymbol{X}, \boldsymbol{X})$ is the initial $\boldsymbol{K}^{\mathrm{NTK}}$ on training data, and $\boldsymbol{K}(\boldsymbol{X}, \boldsymbol{x}) = (\nabla_{\boldsymbol{\theta}} f_{\boldsymbol{\theta}_0}(\boldsymbol{X}))^\top (\nabla_{\boldsymbol{\theta}} f_{\boldsymbol{\theta}_0}(\boldsymbol{x}))$. The asymptotic performance of $\hat{f}(\boldsymbol{x})$ has been analyzed by [1] under the LWR. We view this regime as a *benchmark*: [21, 11] prove that NN through gradient flow is close to lazy training if $h \gg n$; [38] shows NN can go beyond lazy training under a non-proportional regime.

## 3 Case Study for Linear-width NNs

In this section, we investigate a two-layer NN with synthetic data. This setting is promising for future theoretical studies by virtue of RMT. We will showcase the evolution of its spectral properties over training. A two-layer NN in (1) is defined by

$$f_{\boldsymbol{\theta}}(\boldsymbol{x}) := \frac{1}{\sqrt{h}} \sum_{i=1}^h v_i \sigma(\boldsymbol{w}_i^\top \boldsymbol{x}/\sqrt{d}). \tag{7}$$

At initialization, we assume that the first hidden-layer $\boldsymbol{W} = [\boldsymbol{w}_1, \dots, \boldsymbol{w}_h]^\top \in \mathbb{R}^{h \times d}$ is composed of independent standard normal random vectors.

**Assumption 3.1** (Linear-width regime (LWR)). Assume that $\frac{n}{d} \to \gamma_1$ and $\frac{h}{d} \to \gamma_2$ as $n \to \infty$ where the aspect ratios $\gamma_1, \gamma_2 \in (0, \infty)$ are two fixed constants.

LWR stands as a pivotal setting grounded in high-dimensional statistics [1, 61]. It offers valuable insights especially when addressing real-world datasets. This is in contrast to the infinite-width regime, in which we are already in the asymptotic limit for width at first. Hence, LWR is a better approximation of real-world datasets and practical neural networks compared with the infinite-width regime.

**Assumption 3.2** (Activation function). Suppose that the activation function $\sigma(x)$ is nonlinear and $\lambda_\sigma$-Lipschitz with $|\sigma'(x)|, |\sigma''(x)| \le \lambda_\sigma$ for all $x \in \mathbb{R}$. Moreover, $\mathbb{E}[\sigma(z)] = 0$ for $z \sim \mathcal{N}(0, 1)$.

Though the LWR is somewhat impractical, it is still more aligned with deployed models than the infinite-width regime ($h \gg n$, fixed $d$). As a kernel machine, the infinite-width NN has been studied extensively [40, 28, 27, 84]. This infinite-width limit is special, however, as NNs may generally evolve beyond the kernel regime and achieve superior performance [31, 55, 10, 78].

|  | Optimization | Learning rate $\eta$ | $R^2$ score | Test error | Spectra |
|---|---|---|---|---|---|
| Case 1 | GD | 5.0 | 0.63582 | 0.36381 | Invariant Bulk |
| Case 2 | SGD | 0.1 | 0.60605 | 0.36879 | Invariant Bulk |
| Case 3 | SGD | 22.0 | 0.76081 | 0.23791 | Bulk+spike |
| Case 4 | Adam | 0.092 | **0.78829** | **0.21071** | Heavy tail |
| | Lazy regime | | 0.68092 | 0.3185 | |

Table 1: Four models with the same architecture ($n = 2000$, $h = 1500$, $d = 1000$, and $\sigma$ is normalized tanh), but different choices of initial learning rates and optimizers listed in Table 1. The training label noise $\sigma_\varepsilon = 0.3$ and the teacher model is defined by (9) with $\sigma^*$ a normalized *softplus* and $\tau = 0.2$. We observe that simply choosing an optimizer and learning rate can affect the shapes of the final spectra and the performance of the NN, as measured by $R^2$ scores and test errors.

**Assumption 3.3** (Synthetic dataset and teacher model). Training data is $\boldsymbol{X} := [\boldsymbol{x}_1, \dots, \boldsymbol{x}_n] \in \mathbb{R}^{d \times n}$, where $\boldsymbol{x}_i \overset{i.i.d.}{\sim} \mathcal{N}(\boldsymbol{0}, \boldsymbol{I}_d)$. The training labels $\boldsymbol{y} = [y_1, \dots, y_n]$ are defined by $y_i = f^*(\boldsymbol{x}_i) + \varepsilon_i$, for $i \in [n]$, where $f^* : \mathbb{R}^d \to \mathbb{R}$ is the teacher model, and $\varepsilon_i$ is centered sub-Gaussian noise with variance $\sigma_\varepsilon^2$.

One of the simplest nonlinear teacher models we can generate is the single-index model, namely $f^*(\boldsymbol{x}) = \sigma^*(\boldsymbol{x}^\top \boldsymbol{\beta})$ for a fixed vector $\boldsymbol{\beta}$ with $\|\boldsymbol{\beta}\| = 1$ and nonlinear function $\sigma^*$; the hidden feature is simply $\boldsymbol{\beta} \in \mathbb{R}^d$. In general, we can consider a multiple-index model

$$f^*(\boldsymbol{x}) = \frac{1}{k} \sum_{i=1}^k \sigma^*(\boldsymbol{x}^\top \boldsymbol{\beta}_i) \tag{8}$$

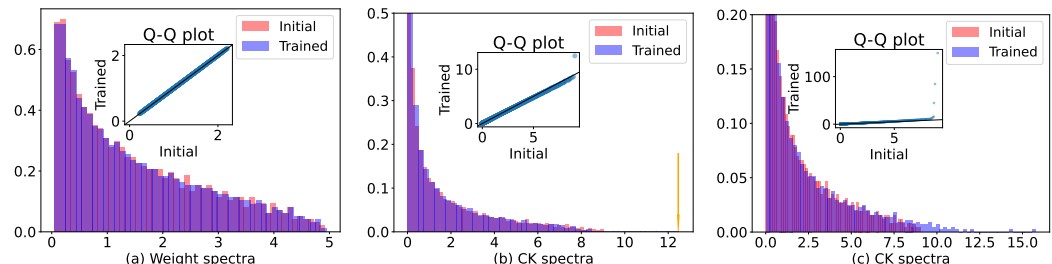

Figure 1: Different spectral behaviors in Table 1: (a) The initial and trained spectra of $\boldsymbol{W}$ in Case 1. The spectrum is invariant based on the Q-Q subplot. (b) The initial and trained spectra of $\boldsymbol{K}^{\text{CK}}$ in Case 3. There is an outlier (orange arrow) in the spectrum after training. (c) The initial and trained spectra of $\boldsymbol{K}^{\text{CK}}$ in Case 4. We refer to Appendix B.2 for other spectra of weight, CK, and NTK matrices in Case 1-4, where analogous phenomena hold for other matrices.

where $\boldsymbol{\beta}_i$ are some orthogonal unit vectors. We will specifically consider a mixture of single-index and quadratic models as our teacher model in this section:

$$f^*(\boldsymbol{x}) = \sigma^*(\boldsymbol{x}^\top \boldsymbol{\beta}) + \frac{\tau}{d} \|\boldsymbol{x}\|^2, \tag{9}$$

for some nonlinear target $\sigma^*$, signal $\boldsymbol{\beta}$ and constant $\tau^3$. Following the above assumptions and constructions, we show different spectral properties (Figure 1) for this two-layer NN using different training procedures (Table 1). Figure 1 exhibits three types of spectra after training: unchanged bulk distribution, bulk with one spike, and heavy tail in spectra. Putting things together, we can see close relationships between the spectra and the generalization of the NN. These different spectral properties actually reveal disparate features learned via different training strategies.

The advantage of this toy model is that we can easily extract the spectral behaviors over training and then compare them with the kernel machine. We use lazy training defined in (6) as our *benchmark* to assist us in determining whether a NN outperforms the associated kernel machine. Table 1 compares the test errors and $R^2$ scores for different optimization cases and the lazy training. By tuning the hyper-parameters, we can find specific situations where NN outperforms the lazy training (see also Figure 10(c) in Appendix B.2).

From Figures 1(a), 12 and 13 in Appendix B.2, one can observe the spectral distributions of the weight, CK and NTK matrices remain invariant and static during training in Cases 1&2, which indicates both cases still belong to the lazy regime. This spectral invariance impedes further feature learning during the training process. The emergence of the outlier in Figures 1(b) and 14 of Appendix B.2, however, shows the improvement over lazy training and potential feature learning via the training process, where the spectra possibly inherit the structures in teacher models (see Section 4.2). Comparing with Case 2, Case 3 of Table 1 suggests the importance of the large learning rate regime for training NNs [52, 64, 55, 15, 3]. As a remark, our spectral results of Case 3 are consistent with the observations in [79] through RMT hypothesis testing, where the majority of trained weight matrices remain random, and the learned feature may be contained in the largest singular value (outlier) and associated vector only. From Figures 1(c) and 16 in Appendix B.2, Case 4 further exhibits more spikes and heavy tails in the trained spectra, which thoroughly goes beyond the realm of initial kernel machine. Notably, this phenomenon is not unique to Adam since heavy tails also occur with AdaGrad in Figure 22 in Appendix B.6. Although all of these cases have the same identical initialization, different methods of optimization eventually lead to various training trajectories and evolutions of the spectra of the weight and kernel matrices. To acquire feature learning, Cases 3&4 cause weights to deviate far from initialization. In the following Section 4, we prove the invariance of the bulk distributions and provide more refined analyses of spikes and heavy tails in terms of feature learning.

---

[3]Here, the norm term of $\boldsymbol{x}$ in (9) is designed to make the teacher model more complicated to be learned. All our empirical results still hold when $\tau = 0$.

# 4 Different Spectral Behaviors in NNs

We now further explore the spectral behaviors in different cases of Table 1 by clarifying how the spectra evolve through different training processes and how this evolution may affect the NN. Following Figure 1, we study the training processes case-by-case: invariant bulk, spikes outside the bulk, and heavy-tailed distribution. Additional experiments are exhibited in Appendix B.

## 4.1 Invariant Bulk Distributions

In Figure 1(a) (also Figures 12 and 13 in Appendix B.2), we observe the bulk distributions of weight and kernel matrices in Cases 1&2 remain globally unchanged (invariant) over the training process. under the LWR, this is also empirically verified by Figures 10(a)&(b) in Appendix B.2. In this section, by investigating the global convergence of GD, we prove this invariant-bulk phenomenon under certain assumptions.

For simplicity, we focus on analyzing the training process of the first-hidden layer with the second layer $\boldsymbol{v}$ fixed. Denote $f_{\boldsymbol{\theta}}(\boldsymbol{X})$ by $f_{\boldsymbol{W}}(\boldsymbol{X})$ in this case. At any time $t \in \mathbb{N}$, consider the gradient steps:

$$\boldsymbol{W}_{t+1} = \boldsymbol{W}_t - \eta \nabla_{\boldsymbol{W}} \mathcal{L}(\boldsymbol{W}_t). \tag{10}$$

Denote the CK and NTK at gradient step $t \in \mathbb{N}$ by $\boldsymbol{K}_t^{\text{CK}} := \frac{1}{h} \sigma(\boldsymbol{W}_t \boldsymbol{X})^\top \sigma(\boldsymbol{W}_t \boldsymbol{X})$, and $\boldsymbol{K}_t^{\text{NTK}} := \frac{1}{d} \boldsymbol{X}^\top \boldsymbol{X} \odot \frac{1}{h} \sigma'\left(\frac{1}{\sqrt{d}} \boldsymbol{W}_t \boldsymbol{X}\right)^\top \text{diag}(\boldsymbol{v}_t)^2 \sigma'\left(\frac{1}{\sqrt{d}} \boldsymbol{W}_t \boldsymbol{X}\right)$ respectively. First, we present an elaborate description of the changes in the weight, CK, and NTK at the *early phase* of the training (after any finite $t$ steps) as follows.

**Lemma 4.1** (Early phase). *Under Assumptions 3.1, 3.2 and 3.3, we further assume that $\|\boldsymbol{v}\|_\infty \leq 1$ and $f^*$ is a $\lambda_\sigma$-Lipschitz function. Given any fixed $t \in \mathbb{N}$ and learning rate $\eta = \Theta(1)$, after $t$ gradient steps, the changes $\frac{1}{\sqrt{d}} \|\boldsymbol{W}_t - \boldsymbol{W}_0\|_F$, $\|\boldsymbol{K}_t^{\text{CK}} - \boldsymbol{K}_0^{\text{CK}}\|_F$, and $\|\boldsymbol{K}_t^{\text{NTK}} - \boldsymbol{K}_0^{\text{NTK}}\|$ are all less than $\frac{C}{n}$, with probability at least $1 - 4n \exp(-cn)$, for some positive constants $c, C > 0$ which only depend on step $t$ and parameters $\eta, \gamma_1, \gamma_2, \lambda_\sigma, \sigma_\varepsilon$.*

Lemma 4.1 shows $\frac{1}{\sqrt{d}} \|\boldsymbol{W}_t - \boldsymbol{W}_0\|$, $\|\boldsymbol{K}_t^{\text{CK}} - \boldsymbol{K}_0^{\text{CK}}\|$, and $\|\boldsymbol{K}_t^{\text{NTK}} - \boldsymbol{K}_0^{\text{NTK}}\|$ are asymptotically vanishing for any fixed time $t$. Therefore, all the eigenvalues/eigenvectors are asymptotically unchanged at the early phase of the training (see Corollary C.3 in Appendix C). Now we aim to analyze the spectra at the end of the training process (10). In this case, although we are unable to show the invariance for each eigenvalue, we can verify the invariance of the limiting bulk distributions for $\boldsymbol{K}_t^{\text{CK}}$ and $\boldsymbol{K}_t^{\text{CK}}$ for all $t$.

By [81, Theorem 2.9], the smallest eigenvalue of $\boldsymbol{K}_0^{\text{NTK}}$ has an asymptotic lower bound:

$$\lambda_{\min}(\boldsymbol{K}_0^{\text{NTK}}) \geq \left(a_\sigma - \sum_{k=0}^{2} \eta_k^2\right) (1 - o_{d, \mathbb{P}}(1)), \tag{11}$$

where $a_\sigma := \mathbb{E}[\sigma'(\xi)^2]$ and $\eta_k$ is the $k$-th Hermite coefficient of $\sigma'$. Hence, we can claim there exists some constant $\alpha > 0$ only dependent on $\sigma$ such that $\lambda_{\min}(\boldsymbol{K}_0^{\text{NTK}}) \geq 4\alpha^2$ with high probability. Note that $\alpha$ is not vanishing since $\sigma$ is nonlinear. With this lower bound, we obtain the following global convergence for (10) and norm control of $\boldsymbol{W}_t$ as $n/d \to \gamma_1$ and $h/d \to \gamma_2$.

**Theorem 4.2** (Global convergence). *Under the same assumptions of Lemma 4.1, we further assume $v_i$'s are independent and centered random variables in the second layer. For any $\eta < \min\{\frac{\alpha^2 n}{2}, \frac{n}{4\lambda_\sigma^2(1+\sqrt{\gamma_1})^2}\}$ and all $t \in \mathbb{N}$, there exists some $\gamma^* > 0$ such that, when $\gamma_2 \geq \gamma^*$, the gradient steps (10) will satisfy*

$$\ell(\boldsymbol{W}_t) \leq \left(1 - \frac{\eta \alpha^2}{2n}\right)^t \ell(\boldsymbol{W}_0), \tag{12}$$

$$\frac{1}{4} \alpha \|\boldsymbol{W}_0 - \boldsymbol{W}_t\|_F + \ell(\boldsymbol{W}_t) \leq \ell(\boldsymbol{W}_0), \tag{13}$$

$$\sum_{t=0}^{\infty} \|\boldsymbol{W}_{t+1} - \boldsymbol{W}_t\|_F \leq \frac{4\ell(\boldsymbol{W}_0)}{\alpha}, \tag{14}$$

*with high probability, as $n/d \to \gamma_1$ and $h/d \to \gamma_2$. Here, training loss $\ell(\boldsymbol{W}) := \|\boldsymbol{y} - f_{\boldsymbol{W}}(\boldsymbol{X})\|$.*

We apply the techniques and results by [69, 70] to obtain Theorem 4.2. Notice that, unlike Lemma 4.1, the largest learning rate we can choose is of order $\Theta(n)$. As a byproduct, the Frobenius norm in (13) implies the following corollary for the invariance of limiting *bulk* distribution.

**Corollary 4.3.** *Under the same assumptions of Theorem 4.2, for all $t \in \mathbb{N}$, with high probability, there exists some constant $R > 0$ such that the changes $\frac{1}{\sqrt{d}} \|\boldsymbol{W}_t - \boldsymbol{W}_0\|_F$, $\|\boldsymbol{K}_t^{CK} - \boldsymbol{K}_0^{CK}\|_F$, and $\|\boldsymbol{K}_t^{NTK} - \boldsymbol{K}_0^{NTK}\|_F$ are all less than $R$ with high probability. This implies the limiting empirical spectra of $\frac{1}{h}\boldsymbol{W}_t^\top \boldsymbol{W}_t$, $\boldsymbol{K}_t^{CK}$ and $\boldsymbol{K}_t^{NTK}$ are the same as the limiting spectra of $\frac{1}{h}\boldsymbol{W}_0^\top \boldsymbol{W}_0$, $\boldsymbol{K}_0^{CK}$ and $\boldsymbol{K}_0^{NTK}$ respectively, almost surely as $n/d \to \gamma_1$ and $h/d \to \gamma_2$.*

Corollary 4.3 is empirically validated by Figure 29 in Appendix C. In addition, based on Figure 31 in Appendix C, one can further extend Corollary 4.3 to the SGD training process. The total path is $O(\sqrt{h})$ in (13) and (14), which is negligible compared with the Frobenius norm of initial weight matrix (which is of order $\Theta(h)$). Thus, gradient descent iterates (10) remain close to initialization and small perturbation of NTK ensures the smallest eigenvalue (11) of NTK is always lower bounded away from zero. Theorem 4.2, however, does not require that the NTK stays unchanged all the time. Moreover, Corollary 4.3 only shows the invariance of the bulk distribution, while the emergence of outliers cannot be excluded from this result. Though we have global convergence in general, we may still move out of the kernel regime. Global convergence cannot explain when a NN in LWR outperforms the kernel regime (Figure 10(c)). Notice that [11, Theorem 5.4] is not directly applicable to show that a NN is still close to lazy training under the LWR. It requires deeper analysis to claim whether a NN still belongs to the kernel regime or already goes beyond in our case. As we will show in Section 4.2, this also relies on the magnitude of the learning rate for GD/SGD.

## 4.2 Alignments for Spiked Models

The outliers appear in the spectra of the trained weight, CK, and NTK matrices (Figure 14 in Appendix B.2) when NNs are optimized with large learning rates. The outlier is especially clear for the NTK matrix (Figure 14(b)). Heuristically, this indicates that the NN is learning the feature from the teacher model $f^*$. In Figure 2(a), Figures 15 and 27 in Appendix B, we empirically exhibit these phenomena for $\boldsymbol{W}$, $\boldsymbol{K}^{\text{CK}}$ and $\boldsymbol{K}^{\text{NTK}}$ respectively through different training processes.

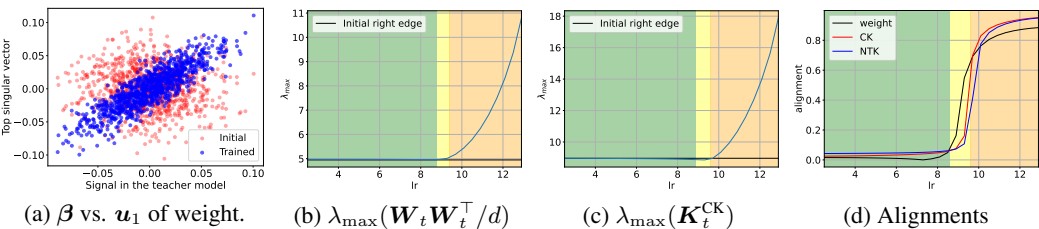

(a) $\boldsymbol{\beta}$ vs. $\boldsymbol{u}_1$ of weight.    (b) $\lambda_{\max}(\boldsymbol{W}_t \boldsymbol{W}_t^\top / d)$    (c) $\lambda_{\max}(\boldsymbol{K}_t^{\text{CK}})$    (d) Alignments

Figure 2: (a) Alignment between teacher feature $\boldsymbol{\beta}$ and first PC of the trained/initial weights in Case 3 of Table 1. (b)-(d) Transitions of $\lambda_{\max}(\boldsymbol{W}_t \boldsymbol{W}_t^\top / d)$, $\lambda_{\max}(\boldsymbol{K}_t^{\text{CK}})$ and alignments ($|\boldsymbol{\beta}^\top \boldsymbol{u}_1| / \|\boldsymbol{\beta}\|$ and $|\boldsymbol{y}^\top \boldsymbol{v}_1| / \|\boldsymbol{y}\|$ where $\boldsymbol{u}_1$ and $\boldsymbol{v}_1$ are the first singular vectors of $\boldsymbol{W}_t$ and either $\boldsymbol{K}_t^{\text{CK}}$ or $\boldsymbol{K}_t^{\text{NTK}}$, respectively) when increasing the learning rate $\eta$ while training the NN with SGD. In the green region, the largest eigenvalues are attached to the bulk (black horizontal lines) and the alignments are weak; in the orange one, outliers become apparent and the alignments become stronger. For different $\eta$, we train the same NN with the same dataset until the training loss is less than $10^{-5}$. Here, "lr" in the $x$-axis represents varying learning rates.

**Spikes of Weight Matrices.** The differences between Cases 2&3 empirically validate the benefits of training with large learning rates [52, 64, 55, 15, 3]. Inspired by [6], we consider the alignment between the leading right singular vector $\boldsymbol{u}_1$ of $\boldsymbol{W}_t$ and the signal $\boldsymbol{\beta}$ in the teacher model defined by (9). For Case 3, a notable alignment appearing in Figure 2(a) after training suggests that $\boldsymbol{W}_t$ is capturing the feature $\boldsymbol{\beta}$ during training. Although this does not ensure NN will entirely beat the optimal kernel lower bound, this alignment reveals a non-negligible feature selection [10] via large-stepsize training. This dynamical alignment along the task-relevant direction may further interpret the generalization of the NN. We also observe similar phenomena for the adaptive optimization in Figure 27 in Appendix B.7.

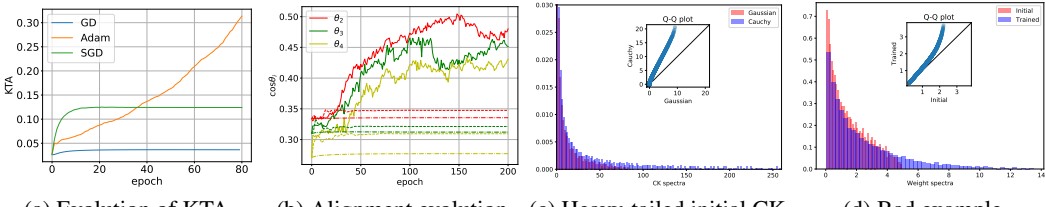

| (a) Evolution of KTA. | (b) Alignment evolution. | (c) Heavy-tailed initial CK. | (d) Bad example. |

Figure 3: (a) Evolution of KTA of CK defined by (15) with respect to training labels for Cases 1, 3&4 in Table 1. We normalize the epoch scales ($x$-axis) for better observations. Heavy-tailed phenomena: (b) Evolutions of PC angles $\theta_i$ between feature subspace $U$ of (8) and top 100 eigenspace of $W_t^\top W_t$ during training with Adam (solid line), SGD (dashed line) and GD (dash-dot). For the first PC $\theta_1$, see Figure 17 in Appendix B.4. (c) The CK spectra at two initializations for $W$: standard Gaussian and Cauchy distributions. (d) Weight spectra at initial and after SGD training. After training the weight reveals a heavy tail, but generalizes not as well as former examples (test loss $1.47504$; $R^2$ score $-0.48$).

**Transitions of the Spike as a Function of Learning Rate.** From Case 2 to Case 3, we observe the emergence of outliers in the trained spectra when increasing the learning rate $\eta$. This indicates a transition of the emergence of the spike outside the bulk distribution. Figure 2, analogously to the well-known BBP transition by Baik, Ben Arous, and Péché in [9] from the RMT community, shows there is a threshold (yellow region) for learning rate: the outliers only appear when $\eta$ exceeds this threshold. We fix the same NN and dataset for all trials of training. The flat black lines in Figures 2(b) and (c) are the right edges of the limiting spectra at initialization. Figure 2(d) records the angles between $\beta$ and the leading eigenvector of $W_t^\top W_t/d$, and $y$ and the leading eigenvectors of $K_t^{\mathrm{CK}}$ and $K_t^{\mathrm{NTK}}$ after training for different $\eta$. Similarly with [10], when $\eta$ is sufficiently large (orange region), we obtain significant alignments which suggest potential feature learning. These transitions of leading eigenvalue and eigenvector alignment have been proved for $W_t$ by [6] for a different scenario[4].

**Spikes of Kernel Matrices.** The alignment of the kernel matrix with the training labels $y$ is defined by [22] by Kernel Target Alignment (KTA) as follows: when kernel $K$ is either CK or NTK,

$$\mathrm{KTA} = \frac{\langle K, y^\top y \rangle}{\|K\|_F \|y\|^2}. \tag{15}$$

Analogously to [10, 5, 78], Figure 3(a) depicts the evolution of KTA of CK in several cases. Based on Figure 2(d), when the spike appears outside the bulk (Case 3), its corresponding (leading) eigenvector $v_1$ of kernel matrix naturally dominates the alignment with $y$ (Figure 15 in Appendix B.2), which is regarded as a kernel rotation during training in [68]. Notice that this is not the common situation in Cases 1&2 of Table 1 (and cf. Figure 11 in Appendix B.2). On the other hand, KTA measures the alignment between $y$ and the full eigenbasis of the kernel. These kernel alignments improve the speed of the convergence of training dynamics but may hurt or boost the generalization of the NNs [68, 78, 10]. Figure 3(a) indicates that Case 4 with heavy-tailed spectra after training has a larger KTA than the other cases. In this case, the emergence of a heavy tail in the spectrum is closely related to a better generalization of the NN and more significant feature learning.

### 4.3 Phenomenon of Heavy-tailed Spectra

Next, we analyze the heavy-tailed spectra of weight and kernel matrices in Figure 1(c). [58, 59] found a strong correlation between the heavy-tailed spectra of trained state-of-the-art models with better generalization (Figure 7 in Appendix B.1). Heavy-tailed spectra can be viewed as an extreme of "bulk+spikes", where a fraction of the eigenvalues move out of the initial bulk. In RMT, heavy-tailed spectra generally appear when the entries of the matrix are highly correlated [59]. This could heuristically explain heavy-tailed phenomena in the spectra since the entries of well-trained $W_t$ should be strongly correlated. Unlike [58, 59], we focus on the heavy-tailed phenomena for both weight and kernel matrices in a simpler model (7) and provide a connection between feature learning and heavy-tailed spectra, which opens an important avenue for further theoretical analysis.

---

[4]We apply NTK parameterization for our neural networks and train both layers until convergence, while [6] considers the mean-field initialization and early stage of training dynamics of GD for the first layer.

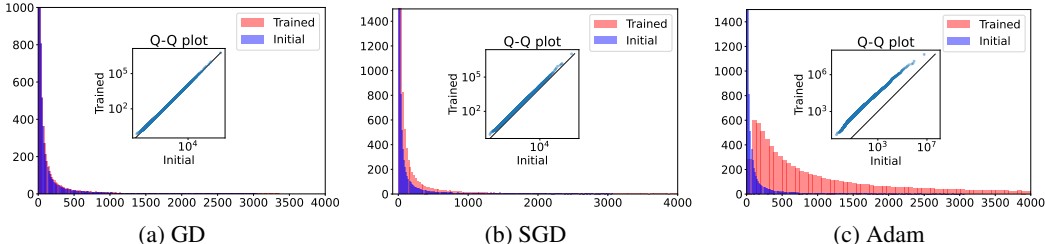

Figure 4: Different NTK spectra for a small-CNN model on CIFAR-2. The subplots are Q-Q plots for the comparison between initial and trained spectra. Test accuracies: (a) 79%, (b) 84%, (c) 86.4%.

**Heavy Tails and Generalization.** We emphasize that heavy tails are not sufficient for good generalization, in general, [60, 63]. Figures 3(c)&(d) exhibit NNs with heavy-tailed weights but in the absence of good performance at initialization. In fact, it is the alignments between the features learned from the heavy-tailed part and the features in the teacher model that finally determine the generalization error of NNs.

More precisely, we provide an example of when heavy tails indicate better generalizations. Consider the multiple-index teacher model (8) with $k = 5$ feature directions $\boldsymbol{\beta}_i$, and train NNs (7) with GD, SGD, and Adam to get invariant bulk, bulk with one spike and heavy tails, respectively, after training. In Figure 3(b), we present the evolutions of the *principle angles* $\theta_i$ between feature subspace $U = \mathrm{span}\{\boldsymbol{\beta}_i\}_{i=1}^k$ and top 100 eigenspace of $\boldsymbol{W}_t^\top \boldsymbol{W}_t$ during different training processes. This eigenspace with respect to the top 100 eigenvalues of $\boldsymbol{W}_t^\top \boldsymbol{W}_t$ corresponds to the heavy-tail part of the spectrum in $\boldsymbol{W}_t^\top \boldsymbol{W}_t$ when training NNs with Adam (solid lines in Figure 3(b)). Comparing with GD and SGD training processes, we observe strong alignments between feature space $U$ and eigenspace w.r.t heavy tails in Adam case in Figure 3(b), which explains why Adam case (NNs with heavy-tailed spectra) generalizes better than the other two cases. For more examples, see Figures 17 and 20 in Appendix B.4. This concludes that NNs with heavy-tailed spectra can generalize better only when the teacher features from data are aligned with the heavy-tailed part of spectra. If the feature dimension in the teacher model is high (i.e. the teacher model is more complicated and intrinsically high-dimensional), then we expect to get a heavy-tailed weight spectrum of well-trained NN where the heavy-tailed part learns all the features in the teacher modes. This example explains why we can use the heavy tails to discriminate well-trained and poorly-trained large models [60, 63, 86].

## 5   Discussions and Future Directions

We empirically investigated how the spectra of $\boldsymbol{W}$, $\boldsymbol{K}^{\mathrm{CK}}$, and $\boldsymbol{K}^{\mathrm{NTK}}$ evolve under the LWR for an idealized student-teacher setting. Our work implies that understanding the relationship between feature learning and training processes requires understanding the evolution of the spectra of both weight and kernel matrices. In particular, we show that different training processes affect the eigenstructure of weight and kernel matrices. Since evolution is sensitive to feature learning, we can link feature learning and different training dynamics by studying the spectra of these matrices.

While synthetic data is easier to analyze theoretically, we also investigate these spectral properties on real-world data and more complicated tasks in the following. In practice, people mainly focus on analyzing spectra of the weight matrices in fully connected layers; we choose to also focus on the spectral properties of general kernel matrices induced by the NNs, which contain abundant information [19, 55, 5, 78].

First, we show the spectra of $\boldsymbol{K}^{\mathrm{NTK}}$ before and after training for binary classification on CIFAR-2 through small CNNs in Figure 4. Similarly with Case 1, Figure 4(a) (especially in the Q-Q subplot) manifests the invariant spectral distribution of NTK through GD training while SGD exhibits a heavier tail in NTK spectrum in Figure 4(b). This phenomenon is more evident when trained by Adam in Figure 4(c) with improved accuracy. Figure 4 suggests that our observations on synthetic data in Section 3 can be extended to real-world data and on more practical architectures. We note that there is a lack of the emergence of spikes after training because spikes already exist in the initial NTK spectrum for this complicated neural architecture on real-world datasets. Figure 4(a) also indicates

that the spectral invariance of NTK through training will impede the feature learning and the NN does not generalize well in this training process.

We also investigate the spectral properties on the pre-trained model, BERT from [26], with fine-tuning on Sentiment140 dataset of tweets[5] from [36]. We fine-tune the BERT model for a binary classifier on Sentiment140 and capture the evolution of CK spectra, rather than the NTK due to the size of BERT, in Figure 5 (see also Figure 7 in Appendix B.1).

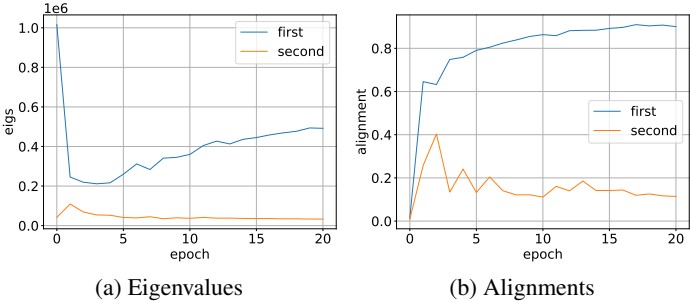

(a) Eigenvalues                (b) Alignments

Figure 5: We use SGD for fine-tuning the BERT model. The training accuracy is $95.90\%$ and the test accuracy is $84\%$. (a) The evolution of first and second eigenvalues of empirical CK during fine-tuning. (b) The alignments of training labels with first and second eigenvectors of CK during fine-tuning. See Figure 7 for the spectra of CK at different epochs.

A heavy-tailed CK spectrum with several spikes already exists in this pre-trained model. Unlike Figure 4 (and cases in Table 1) where the first spike of NTK becomes larger than at random initialization after training, in Figure 5(a), the leading eigenvalue first decreases and then increases. Moreover, similarly to Figure 2(d), our Figure 5(b) shows that the alignment of the first eigenvector of the CK and training labels becomes more apparent through fine-tuning with the leading eigenvalue decrease. Heuristically, this process seems to unlearn the features in the pre-trained model and, remarkably, learn new features on the new dataset in only a few epochs of fine-tuning (see Figure 7in Appendix B.1). We believe that the evolutions of the kernel matrices and some spectral metrics are crucial for understanding feature learning through fine-tuning [82]. A more comprehensive exploration of the evolutionary spectral properties of "foundation models" may help shed further light on these phenomena.

**Limitations.**   Although LWR has garnered significant attention in recent years, e.g., [77, 50, 17, 87, 23], we recognize the limitations inherent in LWR. Our LWR is more realistic compared with infinite-width neural networks and is one of the ways to approximate finite but very large neural networks with very large datasets, but there are more sophisticated regimes for NNs. We leave this for future theoretical work. The NTK parameterization is another limitation of this work. We expect to apply our spectral analysis for other parameterizations of NNs with more real-world datasets. See the discussion at the beginning of Appendix B.

## Acknowledgement

Z.W., A.E., I.D., and T.C. were partially supported by the Mathematics for Artificial Reasoning in Science (MARS) initiative via the Laboratory Directed Research and Development (LDRD) Program at Pacific Northwest National Laboratory (PNNL). A.S. and T.C. were also partially supported by the Statistical Inference Generates kNowledge for Artificial Learners (SIGNAL) program at PNNL. PNNL is a multi-program national laboratory operated for the U.S. Department of Energy (DOE) by Battelle Memorial Institute under Contract No. DE-AC05-76RL0-1830. Z.W. would like to thank Denny Wu and Libin Zhu for their valuable suggestions and comments.

---

[5]https://www.kaggle.com/datasets/kazanova/sentiment140

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

# A Additional Related Work

**Nonlinear Random Matrix Theory and Random Feature Regression.** The limiting spectrum of CK with random input dataset has been investigated by [73, 13]; whereas [57, 30] studied the spectrum of CK with more general input data. This spectrum is actually a deformed Marčenko–Pastur distribution [30], which becomes a deformed semicircular distribution [81] when $h \gg n$. The largest eigenvalue of the CK matrix has been studied in [14], and the spectrum of the NTK was analyzed in [1, 30]. As an application, random feature ridge regression was fully determined by the limiting spectra of CK or NTK: [57, 33, 53, 61, 1]. All of these results belong to LWR.

**Global Convergence of GD for Ultra Wide NNs.** A recent line of work has shown the global convergences of the learning dynamics of gradient-based methods in a certain overparameterized regime, e.g. [28, 27, 69, 70, 66, 54, 75, 18]. We refer to Table 1 of [75] as a summary of these recent results. Most of the theorems in the literature require $h \gg n$, which implies that the NTK is almost static during training, while [70, 66] can consider LWR under some specific assumptions. Recently, [18] established a new criterion for the convergence of GD which results in the global convergence of general NNs with finite width $h$ and $d \geq n$.

**Beyond NTK Regime.** Under the proportional limit, the initial kernel regression can only learn a linear component of the target [35]. Thus, it is reasonable to consider the cases beyond the NTK regime. To this end, [29, 39] considered the dynamics of NTK throughout training while [2, 7] have shown a second-order approximation of NTK, outperforming the initial kernel. In addition, there are many theoretical works analyzing when a NN outperforms the initial kernels in some specific settings: [51] proved a two-layer ReLU NN that is shown to beat any kernel method; [42] verified a two-layer CNN with some simple dataset can outperform the initial NTK for image classifications; [6] showed a NN can escape the kernel regime by only taking one specific large gradient step; [24] showed a specific gradient-based training can even learn polynomials with low-dimensional latent representation.

**Evolution of NTK and Alignment in NNs.** The feature learning can be characterized by the evolution of the kernel during training [31, 68, 55, 5, 56]. Specifically, [55] studied the hard-margin SVM for "after kernels" which are the CK and NTK matrices of trained NNs. One of the effective ways of depicting how the kernels evolve during training is to capture the evolution of kernel alignment [10, 78, 5, 56]. Kernel alignments between kernels and training labels essentially reveal how the NN accelerates training [78]. Also, several papers showed that the top eigenfunctions of the kernel align with the target function learned by the NN [44, 67, 68]. This becomes an efficient way of analyzing how NNs learn features through a particular gradient-based optimization.

**Large Learning Rate Regime.** As mentioned earlier, the large learning rate may contribute to feature learning. The benefits of large-learning-rate training have been studied from different aspects [52, 64, 15, 3]. Specifically, [46] observed that training dynamics with large learning rates differ from the small learning rate regime, where the latter regime exhibits monotone and fast convergence of training loss but may not generalize well on test data. At the early phase of training, [41] showed using lower learning rates may result in finding a region of the loss surface with worse conditioning of kernel and Hessian matrices. In [55], the after kernels of NNs trained with larger learning rates generalize better and stay more stable. [49] raised a "catapult mechanism", where gradient descent dynamics converge to flatter minima for extremely large learning rates. There is a transition as a function of the learning rate, from lazy training to the catapult regime. Section 4.2 illustrates a similar transition in our situations.

**Heavy-tailed Phenomenon.** The heavy-tailed phenomenon has appeared in many places in deep learning theory; [58, 59] observed that many state-of-the-art pre-trained models obtain heavy-tailed weight spectra. More precisely, these spectra have a "5+1" phase transition which relates to different degrees of regularization of the NN. With this heavy-tailed self-regularization theory, [60] further showed how to distinguish well-trained and poorly trained models by a power-law-based approximation. [63] classified trained weight spectra into three types: Marčenko–Pastur law, bulk with (few) outliers, and heavy-tailed spectra. We extend this classification to both weight and kernel matrices in Figure 1. Additionally, similarly to the discussion in 4.3, [63] showed that the difficulty of the classification problem is related to the emergence of heavy-tailed spectra in weight matrices. This

heavy-tailed phenomenon can be used to construct metrics for evaluating the generalization of NNs [60, 86], and early stopping of NNs to avoid over-fitting [63].

# B  Additional Empirical Results

There are different parameterizations for NNs at initialization. The orders of the output of NN are distinct in different cases [21, 25, 85]. This affects the size of stable and non-trivial gradient steps. The distance of trainable parameters from initialization determines whether the NN learns any features from the training data [6, Figure 2]. The performance of networks with different initializations indicates whether the NN belongs to the kernel regime or not [85]. Unlike the NTK parameterization, the *mean-field* parameterization [62, 20] and *maximal update* parameterization [85] tend to be feature learning.

For all NNs in our experiments, we apply a normalized and centered nonlinear activation function such that Assumption 3.2 holds ($\mathbb{E}[\sigma(z)] = 0$ for $z \sim \mathcal{N}(0, 1)$) because we can exclude a large but trivial spike in the initial spectra of kernel matrices. In all architectures of NNs we considered, we remove the bias term of each layer and apply the NTK parameterization (1) with standard Gaussian initialization. Specifically, all entries of $W$ and $v$ in (7) at initialization are i.i.d. standard Gaussian random variables. For all experiments on synthetic datasets, we use standard Gaussian random matrices to generate the training data $X$. In addition, we consider the training label noise defined in Assumption 3.3 as $\varepsilon \sim \mathcal{N}(0, \sigma_\varepsilon^2 \mathbf{I}_n)$.

## B.1  Further Discussions on Real-world Data Experiments

For the first experiment in Section 5, we fix the small-CNN architecture, which is similar to the VGG model, and CIFAR-2 dataset, and vary the methods of optimization of training. Corresponding to Figure 4, the training and test accuracy histories for three cases are shown in Figure 6. Here, in Figure 6(a), we used GD with learning rate $8 \times 10^{-3}$; we used SGD with learning rate $10^{-3}$, batch size 32 and momentum 0.2 in Figure 6(b); Figure 6(c) employs the same learning rate and batch size as 6(b) but employs Adam optimization.

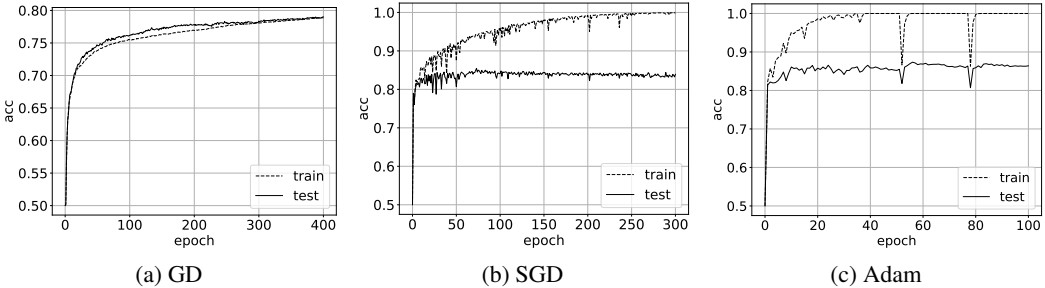

| (a) GD | (b) SGD | (c) Adam |

Figure 6: Training/test accuracy v.s. epochs for small-CNN model on CIFAR-2 with different optimizers.

In the experiment of the transformer language model in Section 5, we fine-tuned the BERT model with SGD for a binary classification on the Sentiment140 dataset. We apply this transformer and fine-tuning to sentiment analysis for social media data. For fine-tuning, the learning rate is 0.003, the batch size is 64 and the momentum is 0.8. The purpose of this experiment is to extract the spectral properties of pre-trained models and the evolution of the CK spectra over fine-tuning. Combining Figure 5, the following Figure 7 exhibits the evolution of the CK spectrum during fine-tuning. Similarly with Case 4 in Table 1, the CK spectrum of this pre-trained model (red histogram in Figure 7) possesses a heavy-tailed distribution, which suggests this transformer has received adequate training. From Figure 7(a) to 7(c), we observe the bulk distribution first shrinks then extends during fine-tuning. This is similar to the evolution of the first eigenvalue of CK in Figure 5(a). Accompanied by this spectra evolution, there is a rapid transformation of the features through fine-tuning, linking the features in the pre-trained model with features in the new dataset. We expect that further spectral analysis will elucidate the feature learning in this kind of transformer [82].

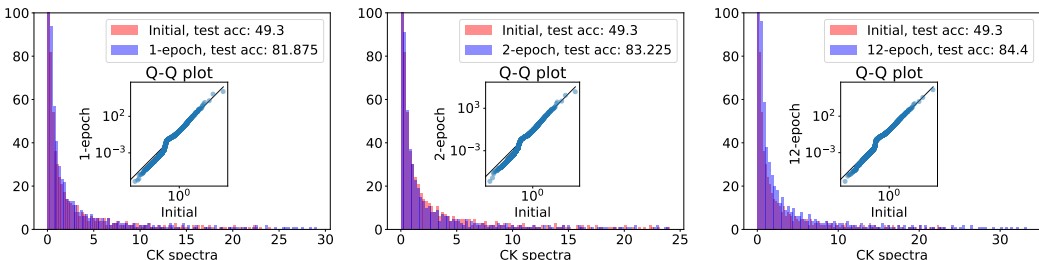

Figure 7: The spectra of CK of the BERT model on Sentiment140 dataset at epoch 1, 2 and 12.

## B.2 Additional Results for Cases in Table 1

To complement the findings in Figure 1 and Section 3, we now present additional results on synthetic data and two-layer NNs. In this section, we will always use the same architecture and dataset as the typical examples in Table 1.

**Norms of the Change.** Based on Figure 1, the trajectories of the weight and kernel matrices are quite different among all cases in Table 1. Hence, for all cases in Table 1, we record the changes in the weight and NTK matrices in both Frobenius norm and operator norm in Figure 8:

$$\frac{1}{\sqrt{d}} \left\| \boldsymbol{W}_0 - \boldsymbol{W}_t \right\|_F, \ \frac{1}{\sqrt{d}} \left\| \boldsymbol{W}_0 - \boldsymbol{W}_t \right\|, \ \left\| \boldsymbol{K}_0^{\mathrm{NTK}} - \boldsymbol{K}_t^{\mathrm{NTK}} \right\|_F, \ \text{and} \ \left\| \boldsymbol{K}_0^{\mathrm{NTK}} - \boldsymbol{K}_t^{\mathrm{NTK}} \right\|,$$

at every epoch $t$ through training. The changes in Figures 8(a) and (b) are much smaller than in the last case. Figure 8(c) has significant changes in both norms after training, which is consistent with the heavy-tailed phenomenon in Figure 1(c). The global optima of the last case is far from the initialization.

Following the settings of Theorem 4.2, in Figure 9, we compute the differences between initial $\boldsymbol{W}_0$ and final $\boldsymbol{W}_s$ in Frobenius norm, operator norm and $2, \infty$-norm. Empirically, Figure 9 shows

$$\frac{1}{\sqrt{d}} \left\| \boldsymbol{W}_0 - \boldsymbol{W}_s \right\|_F, \ \frac{1}{\sqrt{d}} \left\| \boldsymbol{W}_0 - \boldsymbol{W}_s \right\|, \ \frac{1}{\sqrt{d}} \left\| \boldsymbol{W}_0 - \boldsymbol{W}_s \right\|_{2,\infty} = \Theta(1) \tag{16}$$

as $n \to \infty$ with $n/d \to \gamma_1$ and $N/d \to \gamma_2$, where $s$ is the final time for GD. Here, the entry-wise $2\text{-}\infty$ matrix norm is defined as

$$\| \boldsymbol{M} \|_{2,\infty} := \max_{1 \le i \le N} \| \mathbf{m}_i \|,$$

for any matrix $\boldsymbol{M} \in \mathbb{R}^{N \times d}$ with the $i$-th row $\mathbf{m}_i \in \mathbb{R}^d$ and $1 \le i \le N$. Notice that

$$\| \boldsymbol{M} \|_{2,\infty} \le \| \boldsymbol{M} \| \le \| \boldsymbol{M} \|_F. \tag{17}$$

Similar observations for CK and NTK in both Frobenius norm and operator norm are also apparent in Figure 9, which empirically verifies the invariance of the spectra after training. Here, we fix the aspect ratios and let $n$ grow to keep the NNs residing in LWR. For different $n$'s, we repeat the experiments 10 times for average. In each experiment, we train the NN until it converges. As shown in Figure 9(a), the test losses are almost the same for different $n$'s. Figures 9(b)-(d) empirically validate Corollary 4.3. Moreover, the observation that $\frac{1}{\sqrt{d}} \left\| \boldsymbol{W}_0 - \boldsymbol{W}_s \right\|_F$ and $\frac{1}{\sqrt{d}} \left\| \boldsymbol{W}_0 - \boldsymbol{W}_s \right\|$ are $\Theta(1)$ may suggest that $\frac{1}{\sqrt{d}} \left( \boldsymbol{W}_0 - \boldsymbol{W}_s \right)$ is a *low-rank* perturbation. That is, training in LWR may be transferring some low-rank structures to the weight spectrum. This low-rank perturbation can help us better understand the spectral evolution during training. Notice that these norms of the change are different from ultra-wide NN [27, 28]. Similar phenomena can be also observed in Figures 10(a) and (b). Analogous result with different $\sigma$ and $\sigma^*$ is exhibited in Figure 30 in Appendix C. In addition, Figure 10(c) further investigates the cases when NNs can outperform lazy training as defined by (6). In these experiments, we compare the performances of GD, and SGD with small or large learning rates, and lazy training as $n \to \infty$. Each time, we take 10 trials to average. We observe that SGD with a large learning rate (green line) can asymptotically outperform lazy training.

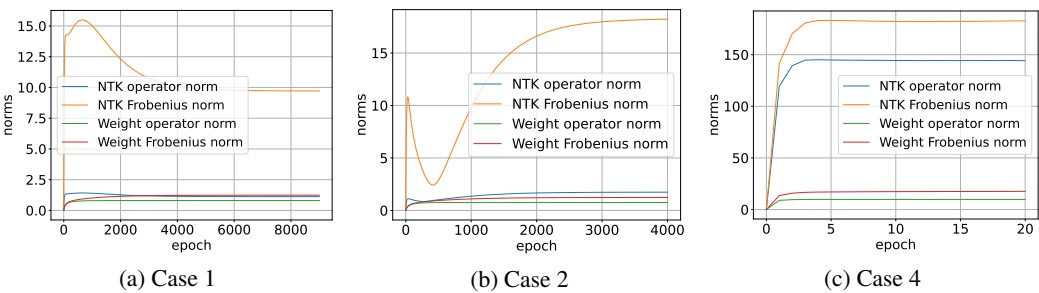

(a) Case 1        (b) Case 2        (c) Case 4

Figure 8: The evolution of the changes in operator/Frobenius norms of the weight/CK/NTK matrices through different training processes. Each case corresponds to the case in Table 1. Case 3 is exhibited in Figure 14(c) below.

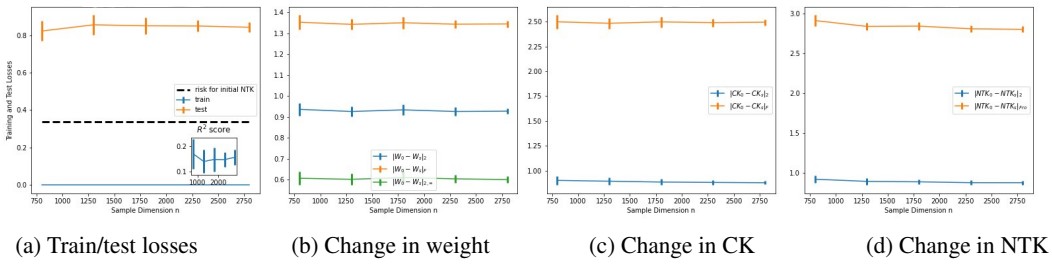

(a) Train/test losses    (b) Change in weight    (c) Change in CK    (d) Change in NTK

Figure 9: Performances of NNs and changes in different norms for weight and kernels, when $d/n = 0.6$ and $N/n = 1.2$ are fixed as $n$ is growing. The activation is normalized $\tanh$ and the teacher model is $f^*(\boldsymbol{x}) = \sigma^*(\boldsymbol{\beta}^\top \boldsymbol{x})$ where $\sigma^*$ is a normalized *softplus*. We average over 10 trials in each case. All these curves are almost flat, which indicates these values are not growing with $\gamma_1$ and $\gamma_2$. Here, in the second figure from the left, we normalized all weights $\boldsymbol{W}$ with $\frac{1}{\sqrt{d}}$ to observe (16).

**Case 1.** Comparing with Figure 15, Figure 11 shows no alignments with training data in GD training. This corresponds to the performances in Table 1. The performance of Case 1 is not as good as the prediction risks in Figure 15, since Figure 11 suggests that no feature learning appears after GD training. Gradient descent requires the weights to converge to some global minima close to initialization, thereby offering no guarantees for lower generalization errors. Next, Figure 12 further presents more results on GD training and indicates more evidence of kernel regime in Case 1. This shows that, from a spectral point of view, the NTK is invariant/static through training. Based on Figures 1(a) and 12, we can empirically verify Corollary C.3 stated in Appendix C. Globally, the spectra of $\boldsymbol{W}$, $\boldsymbol{K}^{\mathrm{CK}}$ and $\boldsymbol{K}^{\mathrm{NTK}}$ are not changing over training as $n/d \to \gamma_1$ and $N/d \to \gamma_2$. The initial spectrum of weight $\boldsymbol{W}_0$ converges to Marčenko–Pastur law; the initial spectrum of NTK under proportional limit has been studied by [1, 30]. Figure 12(c) demonstrates the global convergence for GD under the proportional regime, as proved in Theorem 4.2. We can observe this global convergence even for SGD, Case 2 in Table 1, although we do not have proof for it.

**Case 2.** As a complement, Figure 13 exhibits the spectra of $\boldsymbol{W}$, $\boldsymbol{K}^{\mathrm{CK}}$ and $\boldsymbol{K}^{\mathrm{NTK}}$ for Case 2 in Table 1. The phenomena are similar to Case 1. This observation provides evidence that all results and conjectures in Section 4.1 can be extended to SGD training with sufficiently small learning rates, which is subject to future work. Analogously to Theorem 4.2, we conjecture that the global convergence when training both layers of NN with SGD still holds in this proportional limit. The proof strategy for global convergence, in this case, can again follow [69, 70]. Once we have the invariant global spectra in Corollary 4.3, we can apply the nonlinear RMT [73, 57, 13, 30] to characterize the limiting spectra under LWR.

**Case 3.** Next, in Figures 14 and 15, we present spectral properties for Case 3 in Table 1, where a spike detaches from the bulk after large-step-size training. Notice that Figures 1(b) and 14(a) imply that the bulk spectra for weight and CK remain unchanged over training despite the emergence of spikes. This is not true for NTK by observing Figures 14(b) and (c). The Frobenius norm of NTK changes significantly during training and is not $O(1)$ anymore; the spectra of the first component

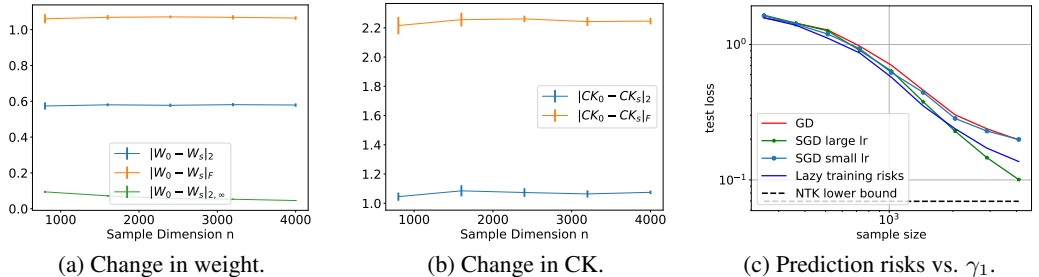

| (a) Change in weight. | (b) Change in CK. | (c) Prediction risks vs. $\gamma_1$. |

Figure 10: (a) Change between initial $\boldsymbol{W}_0$ and final step $\boldsymbol{W}_s$ in operator norm, Frobenius norm and $(2, \infty)$-norm when $d/n = 0.5$, $N/n = 0.8$ are fixed as $n \to \infty$. We train NNs by SGD with $\eta = 2.5$ for 15 trials to average. (b) Change of $\boldsymbol{K}^{\text{CK}}$ in operator norm and Frobenius norm. (c) Prediction risks for lazy training defined by (6), GD with $\eta = \Theta(1)$ (red), SGD with $\eta = \Theta(1)$ (blue dot) and $\eta \propto \gamma_1$ (green), as $\gamma_1 \to \infty$ and $\gamma_2 = 2.5$. The black dashed line stands for the kernel lower bound given by the nonlinear part of the teacher model.

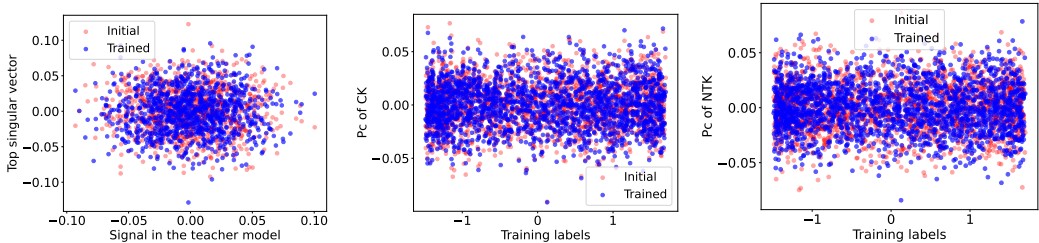

Figure 11: Alignment with leading PCs of trained weight, CK and NTK matrices in Case 1 of Table 1.

of the NTK shrinks after training (Figure 14(b)), which indicates NNs converge to flatter minima. This resembles the catapult phase in [49] for extremely large learning rates. Figure 15(a) shows the convergence rate for SGD in Case 3. Empirically, we observe that the training loss will not monotonically decrease when using a larger learning rate than Case 3, which may be analogous to catapult phases from [49].

**Case 4.** Additional results for Case 4 in Table 1 are shown in Figure 16. Unlike the strong alignments in Case 3 (Figure 15), Figures 16 (e)-(f) do not exhibit strong alignments for the leading singular vectors or eigenvectors. This may be due to the heavy tails present after training, with the other large spikes detaching from the bulks are also important for generalization. A similar phenomenon can be also seen in Figures 23 and 25 in Appendix B.6, where we have comparable performances to Case 4 in Table 1.

Heavy tails are essentially power laws. To measure how "heavy" the spectrum is, [58, 59, 60] provide estimates on the power law of $\boldsymbol{W}$. Consider the empirical spectrum of $\boldsymbol{W}$ as $\rho(x) \sim x^{-\alpha}$ for large $x$ and some positive constant $\alpha$. The spectrum with a heavier tail has a smaller value of $\alpha$. Figure 17(a) shows how $\alpha$ evolves through training in Case 4 of Table 1. As $\alpha$ decreases, a heavy tail in the spectrum of the weight matrix emerges in Figure 1(c). In Figure 17(a) we introduce two more metrics to show this evolution: Weighted Alpha $\hat{\alpha} := \alpha \lambda_1$ and Log $\alpha$-norm $\log \left( \sum_{i=1}^{N} \lambda_i^{\alpha} \right)$ where $\lambda_N \leq \ldots \leq \lambda_1$ are the eigenvalues of $\boldsymbol{W}\boldsymbol{W}^\top$. Remarkably, Figure 17(a) indicates the spectra change dramatically at the early stage of training, which matches the observation from fine-tuning via BERT on real-world data in Figure 5 in Section 5. These metrics are applied to measure the tails in pre-trained models [60].

## B.3 Additional Results for the Emergence of A Spike

As a complement to section 4.2, in Figure 18, we show the training dynamics for SGD training with a larger learning rate in the example of Figure 2(b-d). Here we consider $\eta = 24$ which belongs to the orange region in Figure 2(b-d), where the spike and eigenvector alignment emerge. Figure 18 presents the details of the training dynamics of the NN in this case: the largest eigenvalues of CK and

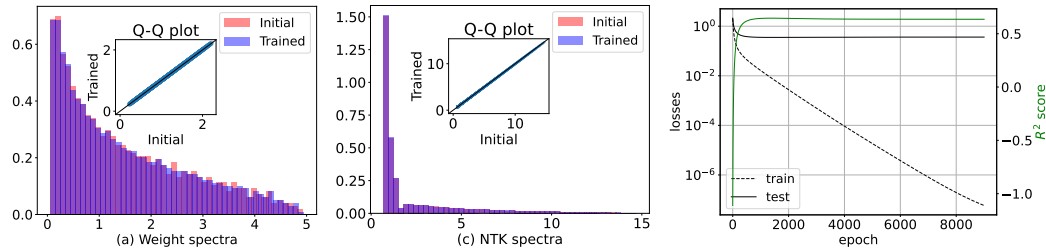

Figure 12: Performances of Case 1 in Table 1: (a) The initial and trained spectra of the first-hidden layer $\boldsymbol{W}$. (c) The initial and trained spectra of empirical NTK matrix defined by (5). Q-Q subplot shows these two spectra are almost the same. Training and test losses vs. epochs for GD (right).

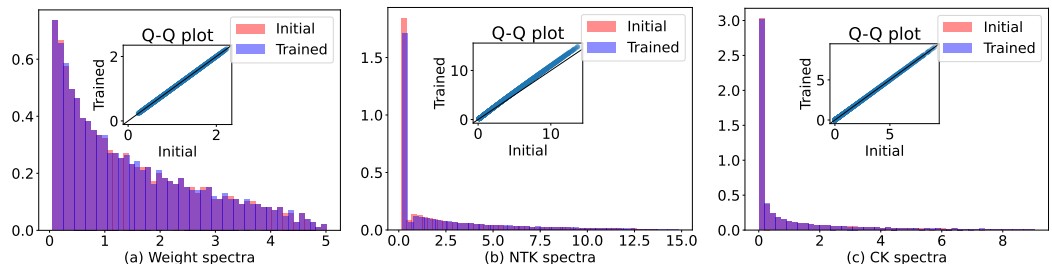

Figure 13: Spectral properties for Case 2 in Table 1: (a) The initial and trained spectra of the first-hidden layer $\boldsymbol{W}$. (b) The initial and trained spectra of empirical NTK are defined by (5). (c) The initial and trained spectra of empirical CK defined by (4).

NTK both increase and the losses first increase and then drop. In Figure 19, we empirically justify that the phase transitions we presented in section 4.2 for SGD can be also extended to full-batch GD cases. We can also observe phase transitions for test losses and $R^2$ scores when we are gradually increasing the learning rates. Parallel to these, a spike also appears outside the bulk distribution, which corresponds to feature alignments in Figure 19(c)&(f).

## B.4   Multiple-index Examples for Heavy-Tailed Spectra in Section 4.3

Figures 17(b) and (c) are additional results for Figure 3(b) in Section 4.3. In this experiment, we consider $\sigma = $ ReLU, $n = 5000$, $h = 2500$ and $d = 1000$ for NN (1). Comparing with the teacher model (9) used in Table 1, we employ the multiple-index teacher model (8) with $k = 5$ and $\sigma^* = \sigma$. We trained this student-teacher model using GD ($\eta = 15$), SGD ($\eta = 7.25$ and batch size 8), and Adam ($\eta = 0.007$ and batch size 16) for training this NN, respectively. Similarly with Figure 1, correspondingly, we observe invariant spectrum, bulk with one spike, and heavy tails after training respectively. Heuristically, to learn this $f^*$, the weight $\boldsymbol{W}$ of NN should gradually align with the feature space $U$ spanned by $\boldsymbol{\beta}_i$'s. Hence, to study feature learning, we can apply principle angles to measure the alignment between $\boldsymbol{W}$ and $U$. Consider the eigen-decomposition of $\boldsymbol{W}_t^\top \boldsymbol{W}_t = \sum_{i=1}^d \lambda_i \boldsymbol{v}_i \boldsymbol{v}_i^\top$ with $\lambda_1 \geq \lambda_2 \geq \ldots \geq \lambda_d$. Figure 3(b) shows the heavy-tailed part (the eigenspace $E := \text{span}\{\boldsymbol{v}_i\}_{i=1}^{100}$) is aligned with $U$ after training, which shows how features are learned in the heavy-tailed spectra. Remarkably, the test errors for training processes with SGD and Adam are even smaller than $\|\mathrm{P}_{>1} f^*\|^2$ and $\|\mathrm{P}_{>2} f^*\|^2$, where $\mathrm{P}_{>1}$ denotes the orthogonal projection onto the nonlinear part of the function w.r.t. Gaussian measure. Thus, we experimentally showed that NNs with heavy-tailed spectra can obtain feature learning and generalize better than the other two cases. Another example is exhibited in Figure 20. In this case, $k = 5$ and there are five leading outlier eigenvalues in the spectrum of the trained weight matrix, along with a heavy-tailed bulk. Interestingly, Figure 20 justifies that the eigenspace of these five leading outliers is strongly aligned with features $\boldsymbol{\beta}_i$ for $1 \leq i \leq 5$. This indicates that heavy-tailed spectra with large spikes may have a correlation with feature learning and good generalizations.

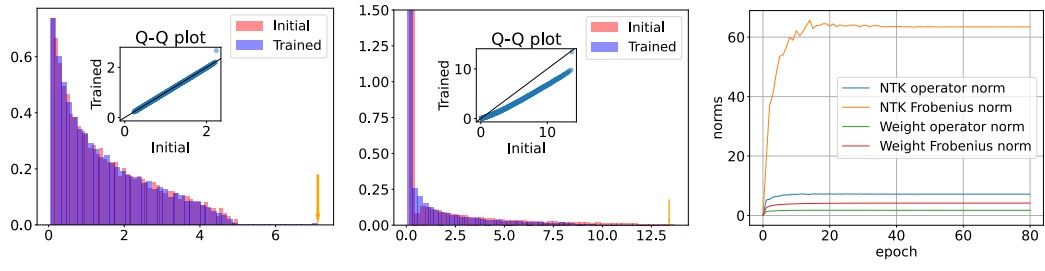

(a) Spectra of weights.  (b) Spectra of the first part in NTKs.  (c) Norms of change vs. epochs.

Figure 14: Additional performance for Case 3 in Table 1: (a) The initial and trained weight spectra. Notice that there is one outlier after training, while the bulk remains invariant. This is analogous to the behavior of CK spectra in Figure 1(b). (b) The spectra of the first part in (5) at initialization and after training. The orange arrow points out the outlier of the spectrum. (c) The changes $\|\boldsymbol{W}_t - \boldsymbol{W}_0\|$, $\|\boldsymbol{W}_t - \boldsymbol{W}_0\|_F$, $\left\|\boldsymbol{K}_t^{\mathrm{NTK}} - \boldsymbol{K}_0^{\mathrm{NTK}}\right\|$ and $\left\|\boldsymbol{K}_t^{\mathrm{NTK}} - \boldsymbol{K}_0^{\mathrm{NTK}}\right\|_F$ at each epoch $t$ throughout the training process.

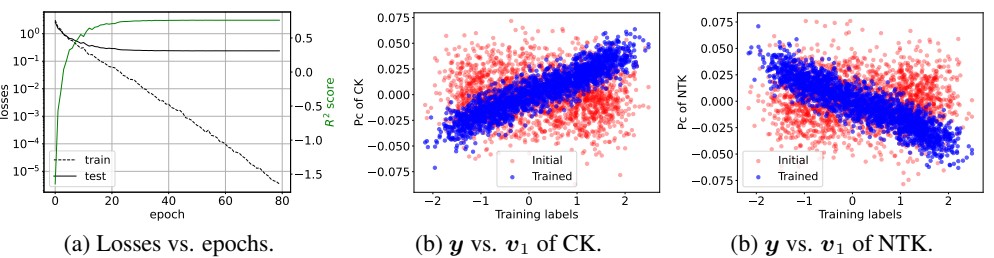

(a) Losses vs. epochs.  (b) $\boldsymbol{y}$ vs. $\boldsymbol{v}_1$ of CK.  (b) $\boldsymbol{y}$ vs. $\boldsymbol{v}_1$ of NTK.

Figure 15: (a) Training/test losses and $R^2$ scores at each epoch of training in Case 3. (b) Alignment between training labels $\boldsymbol{y}$ and first PC of trained/initial CK. (c) Alignment between training labels $\boldsymbol{y}$ and the first PC of trained/initial NTK. We use the same setting in Case 3 of Table 1. There are strong alignments between kernels and training labels as stated in Section 4.2.

## B.5 Training Only the First Hidden Layer

We now present additional results when only training the first layer of NN with Adam. This result resembles Figure 29 in the next section. In section 4.1, Corollary 4.3 shows that under LWR with sufficiently large width $h$, the limiting spectra of $\frac{1}{h}\boldsymbol{W}_t^{\top}\boldsymbol{W}_t$, $\boldsymbol{K}_t^{\mathrm{CK}}$ and $\boldsymbol{K}_t^{\mathrm{NTK}}$ are essentially the same as those of the corresponding initial matrices if we train only the first layer with GD (see (10)). Figure 21 further investigates these phenomena when training only the first layer with Adam. In particular, the Q-Q plots show the invariant spectra of weight, CK, and NTK matrices even when the training loss is approaching zero. Here, in Figure 21(c), we only consider the first component of NTK since the gradient is only taken with respect to $\boldsymbol{W}_t$. Another observation is that the smallest eigenvalues of the initial and trained NTK are both bounded away from zero. This is crucial for the proof of the global convergence as shown in Appendix C.

## B.6 Adaptive Gradients

Inspired by Case 4 in Table 1, we show the spectral performances of adaptive gradient (AdaGrad) in Figures 22 and 23. The performance of this method matches Case 4 in Table 1, where we can also easily observe heavy tails and detaching spikes after training, especially in Q-Q subplots. This suggests that adaptive optimization is more likely to yield heavy-tailed distributions in trained NNs. Besides, analogously to Figure 16, there is no strong alignment in the single leading PC of the weight or kernel matrices after training in Figure 23.

## B.7 Different Global Minima and Alignments

In this section, to distinguish the different alignments in Case 3&4 of Table 1, we introduce the following two simulations with slightly different optimizers to get quite different spectra and alignments among leading PCs after training. In the first experiment, Figures 24 and 25, we first take Adam with large stepsizes for a few steps and then use small-stepsize SGD for convergence. In this scenario, we

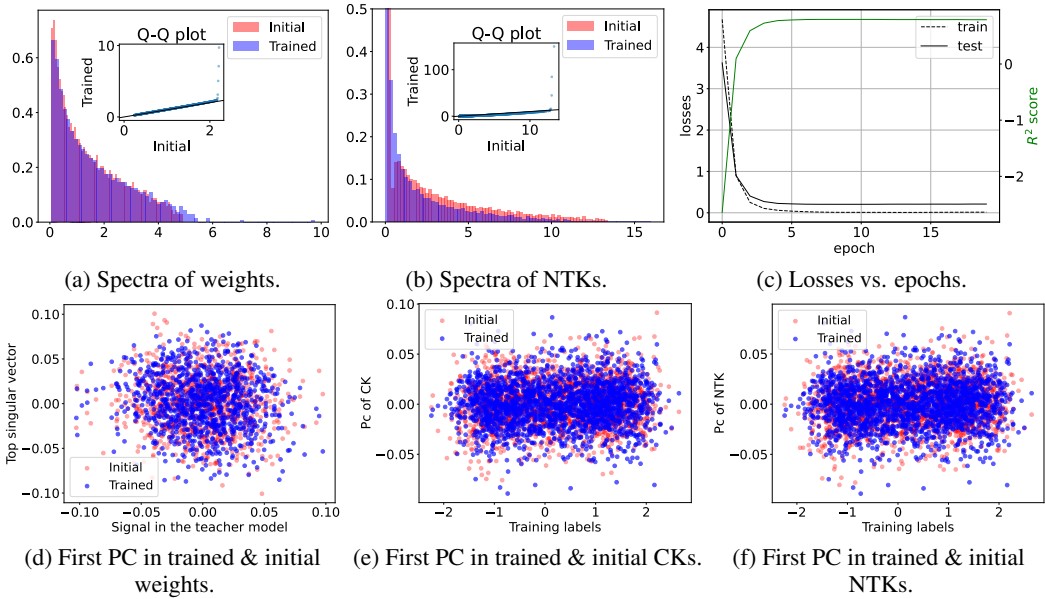

(a) Spectra of weights.

(b) Spectra of NTKs.

(c) Losses vs. epochs.

(d) First PC in trained & initial weights.

(e) First PC in trained & initial CKs.

(f) First PC in trained & initial NTKs.

Figure 16: Additional performance for Case 4 in Table 1: (a) The initial and trained weight spectra. Notice that there are several outliers after training, while the bulk has a heavier tail. (b) The spectra of the NTK (5) at initialization and after training. (c) The test/training losses and $R^2$ score (green line) at each epoch $t$ throughout training process. (d) Alignment between the leading PC of the weight matrix and the signal $\boldsymbol{\beta}$ in the teacher model before (red) and after (blue) training. (e) Alignment between the leading PC of the CK matrix and the training labels $\boldsymbol{y}$ before/after training. (f) Alignment between the leading PC of the NTK matrix and $\boldsymbol{y}$ before/after training.

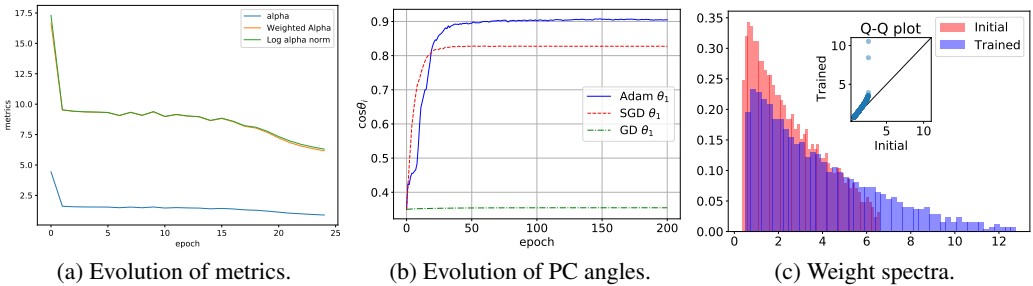

(a) Evolution of metrics.

(b) Evolution of PC angles.

(c) Weight spectra.

Figure 17: (a) The evolution of power $\alpha$, weighted Alpha and Log $\alpha$-norm (several metrics of power law tails; see [60]) during the training process in Case 4 of Table 1. (b) Evolutions of the first PC angle $\theta_1$ between *feature subspace* $U = \text{span}\{\boldsymbol{\beta}_i\}_{i=1}^{k}$ of the multiple-index model (8) and the *eigenspace* spanned by top 100 of eigenvectors of $\boldsymbol{W}_t^\top \boldsymbol{W}_t$ during training with Adam (blue solid line), SGD ( red dashed line) and GD (green dash-dot). The final test error is 0.33865 and the $R^2$ score is -0.71065 for GD. The test error is 0.10814 and the $R^2$ score is 0.45373 for SGD, where one spike emerges in the weight spectrum after training. The test error is 0.08672 and the $R^2$ score is 0.56195 for Adam. (c) Initial and trained spectra for weight matrices when training with Adam (blue solid line in (b)). Heavy tail emerges in this case.

can get heavy-tailed distributions after training, and the phenomena are essentially the same as Case 4 of Table 1. There is no strong alignment for the first leading eigenvector, while useful features may be learned by a few top eigenvectors in the heavy-tailed spectra after training. In the second experiment, we directly apply Adam with a small initial learning rate for training. In contrast, the results of this case, presented in Figures 26 and 27, are similar to Case 3 in Table 1. The test loss and $R^2$ score are close to previous examples, though slightly worse. As explained in the previous section, because there is only one spike appearing outside the bulk after training in the second case, the leading PC is highly aligned with the training dataset structure after training and the feature learning mainly stems from the outliers in this situation. This interprets the strongly anisotropic structures in trained spectra of NNs in the second case [67, 68, 78]. These two different spectral properties reveal significant differences between the global minima of these two training processes and different evolutions of the

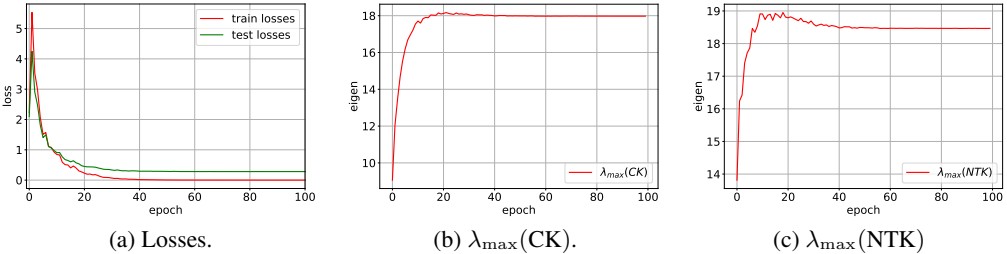

(a) Losses.      (b) $\lambda_{\max}(\text{CK})$.      (c) $\lambda_{\max}(\text{NTK})$

Figure 18: The training dynamic when training neural networks with SGD and learning rate 24 in the example of Figure 2(b-c). The learning rate we chose here is above the threshold we showed in Figure 2(b-c). We use the same architecture, dataset, and teacher model as in Section 3 of our paper. The batch size is 32. (a) The evolution of the training and test errors during training. (b) The evolution of the largest eigenvalue of the CK matrix. (c) The evolution of the largest eigenvalue of the NTK matrix. This regime corresponds to the catapult phenomenon [49].

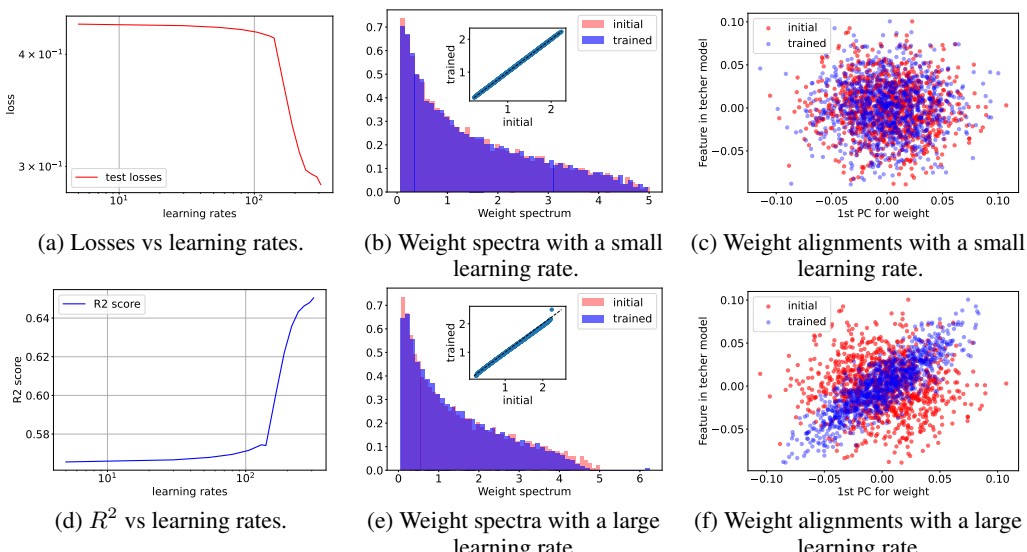

(a) Losses vs learning rates.    (b) Weight spectra with a small learning rate.    (c) Weight alignments with a small learning rate.

(d) $R^2$ vs learning rates.    (e) Weight spectra with a large learning rate.    (f) Weight alignments with a large learning rate.

Figure 19: Grid search for different learning rates when training NNs with full-batch GD in the same setting as Case 1 in Table 1. (a) Final test losses when varying learning rates. (d) $R^2$ scores when varying learning rates. For all these learning rates, we did not observe heavy-tailed spectra. (b-c) present the spectral behaviors for the smallest learning rate we used in (a)&(d). (e-f) present the spectral behaviors for the largest learning rate we can use which still ensures the convergence of GD. In this case, analogously to the SGD case in Section 4.2, we observe an outlier in the trained weight matrix and strong alignment with the spike.

spectra in NNs. Remarkably, based on the different spectral behaviors in trained weight and kernel matrices, these two experiments exhibit disparate features learned by distinct training procedures. Hence, analyzing the spectral properties in trained kernel matrices is beneficial for clarifying what features our NNs have learned during the training processes.

## C   Proofs of Results in Section 4.1

### C.1   GD Analysis at Early Phase

From (10), the GD process with learning rate $\eta > 0$ can be written by

$$\boldsymbol{W}_{t+1} = \boldsymbol{W}_t + \eta \cdot \boldsymbol{G}_t, \text{ where} \tag{18}$$

$$\boldsymbol{G}_t = \frac{1}{n\sqrt{dh}} \left[ \left( \boldsymbol{v} \left( \boldsymbol{y} - \frac{1}{\sqrt{h}} \boldsymbol{v}^\top \sigma(\boldsymbol{W}_t \boldsymbol{X}/\sqrt{d}) \right) \right) \odot \sigma'(\boldsymbol{W}_t \boldsymbol{X}/\sqrt{d}) \right] \boldsymbol{X}^\top, \tag{19}$$

for $t \in \mathbb{N}$, where $\boldsymbol{y} \in \mathbb{R}^{1\times n}$. Following [6, Appendix B], in this section we prove the control for gradient step $\boldsymbol{G}_t$. For simplicity, denote $f_t(\boldsymbol{X}) := f_{\boldsymbol{\theta}_t}(\boldsymbol{X}) = \frac{1}{\sqrt{h}} \boldsymbol{v}^\top \sigma(\boldsymbol{W}_t \boldsymbol{X}/\sqrt{d})$ for $t \in \mathbb{N}$.

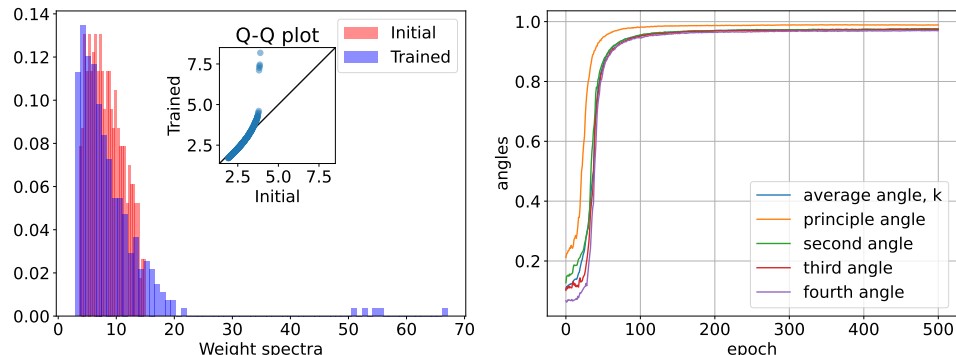

Figure 20: (Left) Initial and trained spectra for weight matrices when training with Adam. Five leading spikes emerge in this case. (Right) Evolutions of the angles between the first four PCs of $\boldsymbol{W}_t^\top \boldsymbol{W}_t$ and *feature subspace* $U = \mathrm{span}\{\boldsymbol{\beta}_i\}_{i=1}^k$ of the multiple-index model (8) during training with Adam. Here $k = 5$. The final test error is 0.33865 and $R^2$ score is -0.71065 for GD. The test error is 0.01681 and $R^2$ score is 0.9154 for Adam.

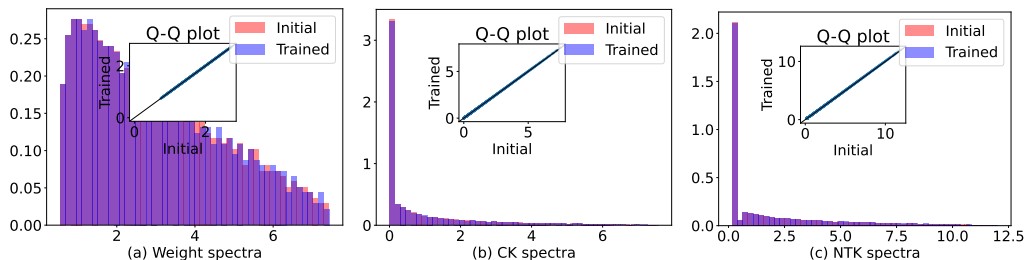

Figure 21: Additional spectral performance when training the NN by Adam with small learning rate $\eta = 0.001$, where $n = 2000, h = 3000, d = 1000, \sigma_\varepsilon = 0.3$ and batch size is 100. In this simulation, we only train the first hidden layer $\boldsymbol{W}_t$. The activation function $\sigma$ is a normalized softplus and the target function is a normalized $\tanh$. The final test loss is 0.36219 and $R^2$ score is around 63.70%.

**Lemma C.1.** *Under the same assumptions as in Lemma 4.1, we have*

$$\mathbb{P}\left( \left\| \sigma(\boldsymbol{W}_0 \boldsymbol{X}/\sqrt{d}) \right\| \geq C\sqrt{n} \right) \leq 2e^{-cn},$$

$$\mathbb{P}\left( \|\boldsymbol{y}\| \geq C\sqrt{n} \right) \leq 2e^{-cn},$$

*for some constants $C, c > 0$ only depending on $\sigma_\varepsilon$, $\lambda_\sigma$, $\gamma_1$, and $\gamma_2$.*

*Proof.* Due to [30, Lemma D.4.], we can directly obtain that

$$\mathbb{P}\left( \left\| \sigma(\boldsymbol{W}_0 \boldsymbol{X}/\sqrt{d}) \right\| \geq C'(\sqrt{n} + \sqrt{h})\sqrt{\frac{h}{d}} \right) \leq 2e^{-cn}.$$

Here we use the fact that both $\boldsymbol{W}_0$ and $\boldsymbol{X}$ are i.i.d. Gaussian random matrices. Then by Assumption 3.1, we conclude that we control $\sigma(\boldsymbol{W}_0 \boldsymbol{X}/\sqrt{d})$. Recall that Assumption 3.3 implies that $\boldsymbol{y} = f^*(\boldsymbol{X}) + \varepsilon$. Hence, by Lipschitz Gaussian concentration inequality [80, Theorem 5.2.2], each entry of $f^*(\boldsymbol{X})$ has independent sub-Gaussian coordinates, whence we can get $\|f^*(\boldsymbol{X})\| \leq C\sqrt{n}$ with probability at least $1 - 2ne^{-cn}$ for some constants $c, C > 0$. On the other hand, $[\varepsilon]_i = \varepsilon_i$ are i.i.d. centered sub-Gaussian noises with variance $\sigma_\varepsilon^2$. By [80, Theorem 3.1.1], we have

$$\mathbb{P}\left( \|\varepsilon\| \leq 2\sigma_\varepsilon\sqrt{n} \right) \geq 1 - 2\exp\left( -\frac{cn}{K^4} \right),$$

where the constant $K$ is the sub-Gaussian norm defined by $K = \max_i \|\varepsilon_i\|_{\psi_2}$. Hence, combining all things together, we obtain the second inequality of this lemma.

$\square$

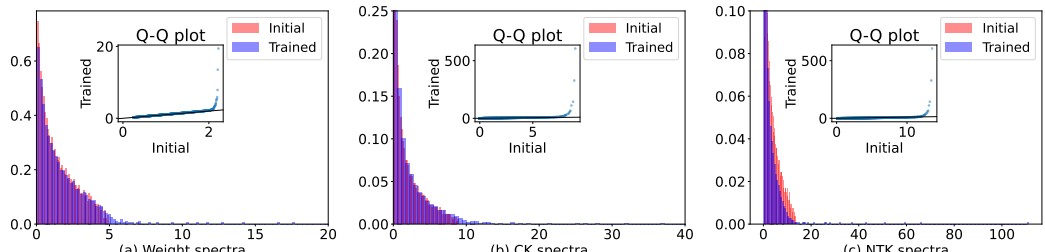

Figure 22: Additional performance for AdaGrad with learning rate $\eta = 0.5$, where $n = 2000, h = 1500, d = 1000, \sigma_\varepsilon = 0.3$ and small batch size is 8. Activation $\sigma$ is normalized softplus and target is normalized $\tanh$. The final test loss is 0.23555 and $R^2$ score is around 0.76249. The black lines in the Q-Q subplots are the line of $y = x$.

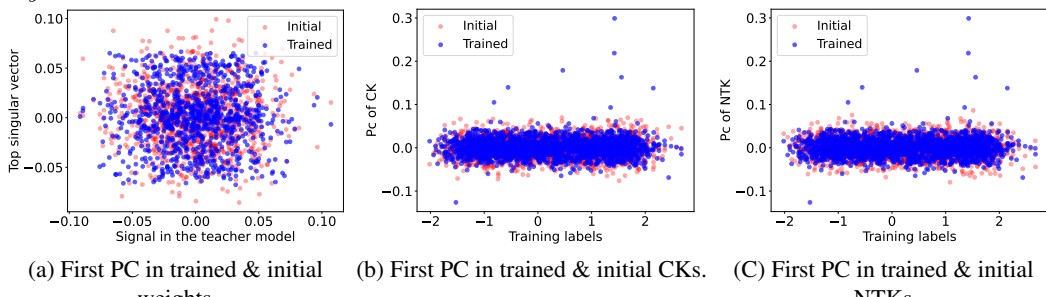

(a) First PC in trained & initial weights.

(b) First PC in trained & initial CKs.

(C) First PC in trained & initial NTKs.

Figure 23: Alignment between the leading PC of the weight/kernel matrices and the signal $\boldsymbol{\beta}$ or the training labels $\boldsymbol{y}$ before/after training for experiment in Figure 22. Analogously to Case 4 in Appendix B.2, there is no strong alignment in the leading component of weight/kernel matrices.

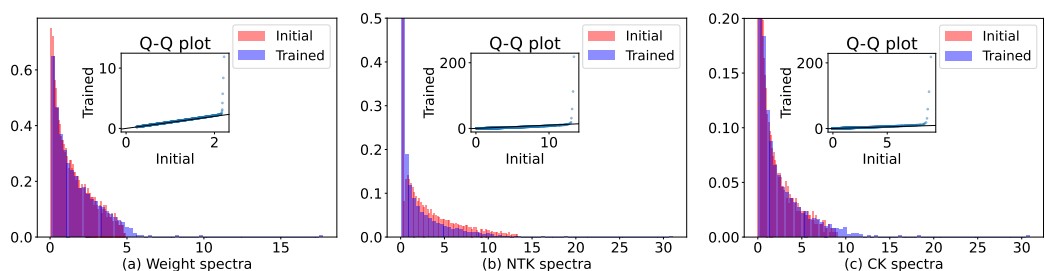

Figure 24: Additional performance for Adam with learning rate $\eta = 0.09$ and 4 epochs, then SGD with learning rate $\eta = 5 \times 10^{-4}$ and 100 epochs, where $n = 2000, h = 1500, d = 1000, \sigma_\varepsilon = 0.3$ and the batch size is 32. We train the NN until the training loss is less than $10^{-10}$. The activation $\sigma$ is normalized softplus and the target is normalized $\tanh$. The final test loss is 0.22511 and $R^2$ score is around 0.77462.

**Lemma C.2.** *Under the assumptions of Lemma 4.1, given any fixed $t \in \mathbb{N}$ and learning rate $\eta = \Theta(1)$, the weight matrix after $t$ gradient steps $\boldsymbol{W}_t$ defined in (18) satisfies*

$$\mathbb{P}\left(\|\boldsymbol{W}_t - \boldsymbol{W}_0\|_F \geq \frac{C}{\sqrt{n}}\right) \leq \exp\left(-cn\right), \qquad (20)$$

*for some positive constants $c, C > 0$ only depending on $t, \eta, \sigma_\varepsilon, \lambda_\sigma, \gamma_1$ and $\gamma_2$.*

*Proof.* Denote $\sigma_\perp(x) = \sigma(x) - \mu_1 x$ which is the nonlinear part of $\sigma$ and $\mu_1 = \mathbb{E}[z\sigma(z)]$. Thus, $\mathbb{E}[\sigma_\perp(z)z] = 0$ for $z \sim \mathcal{N}(0, 1)$. Based on this, we can further decompose the gradient $\boldsymbol{G}_t$ into

$$\boldsymbol{G}_t = \underbrace{\frac{\mu_1}{n\sqrt{dh}}\boldsymbol{v}\left(\boldsymbol{y} - f_t(\boldsymbol{X})\right)\boldsymbol{X}^\top}_{\boldsymbol{A}^t} + \underbrace{\frac{1}{n\sqrt{dh}}\left(\boldsymbol{v}\left(\boldsymbol{y} - f_t(\boldsymbol{X})\right) \odot \sigma'_\perp(\boldsymbol{W}_t\boldsymbol{X}/\sqrt{d})\right)\boldsymbol{X}^\top}_{\boldsymbol{B}^t}. \qquad (21)$$

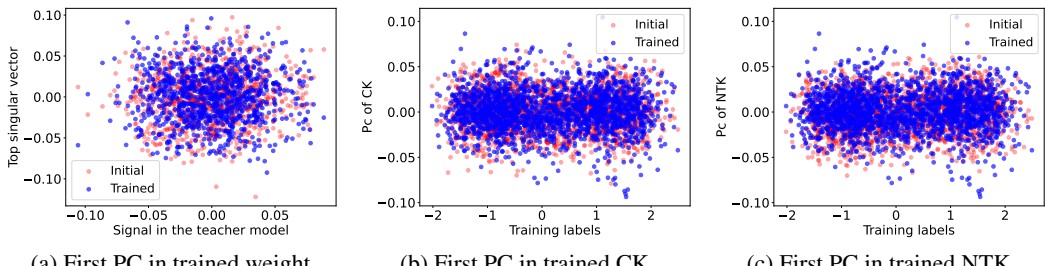

(a) First PC in trained weight.    (b) First PC in trained CK.    (c) First PC in trained NTK.

Figure 25: Alignment between the leading PC of the weight/kernel matrices and the signal $\boldsymbol{\beta}$ or the training labels $\boldsymbol{y}$ before/after training for the experiment in Figure 24. This is analogous to Case 4 in Appendix B.2.

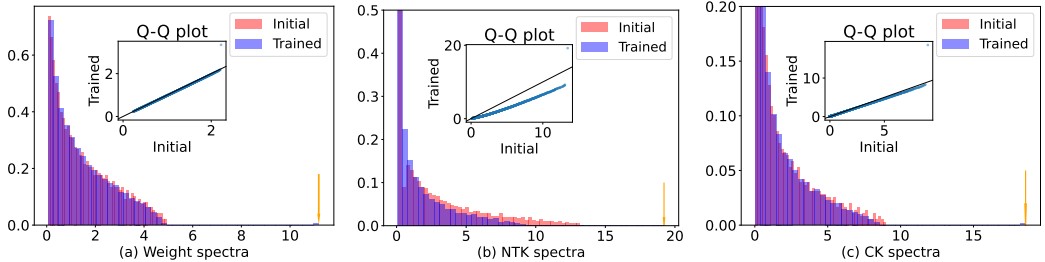

(a) Weight spectra    (b) NTK spectra    (c) CK spectra

Figure 26: Additional performance for Adam with learning rate $\eta = 0.002$ and 700 epochs, where $n = 2000, h = 1500, d = 1000, \sigma_\varepsilon = 0.3$ and batch size is 64. The activation $\sigma$ is normalized softplus and the target is normalized $\tanh$. The final test loss is 0.23954 and $R^2$ score is around 0.76027. The orange arrows show the positions of the outliers. Spectra behaviors in this case differ from Figure 24.

At first, consider $t = 0$ and bound the spectral norm of $\boldsymbol{W}_1$. By assumption, we know $\|\boldsymbol{v}\| \leq \sqrt{h}$. Due to Corollary 7.3.3 in [80], we have

$$\mathbb{P}\left( \frac{1}{\sqrt{d}}\|\boldsymbol{X}\| \geq 2\left(1 + \sqrt{\frac{n}{d}}\right) \right) \leq 2\exp\left(-cn\right). \tag{22}$$

Therefore, by (21), we can control $\boldsymbol{A}^0$ and $\boldsymbol{B}^0$ separately. Notice that, as a rank-one matrix,

$$\left\|\boldsymbol{A}^0\right\| = \left\|\boldsymbol{A}^0\right\|_F \leq \frac{\mu_1}{\sqrt{n}}\frac{\|\boldsymbol{X}\|}{\sqrt{d}}\frac{1}{\sqrt{n}}(\|\boldsymbol{y}\| + \|f_0(\boldsymbol{X})\|)\frac{\|\boldsymbol{v}\|}{\sqrt{h}}$$
$$\leq \frac{\mu_1}{\sqrt{n}}\frac{\|\boldsymbol{X}\|}{\sqrt{d}}\frac{\|\boldsymbol{v}\|}{\sqrt{h}}\frac{1}{\sqrt{n}}\left(\|\boldsymbol{y}\| + \frac{\|\boldsymbol{v}\|}{\sqrt{h}}\left\|\sigma(\boldsymbol{W}_0\boldsymbol{X}/\sqrt{d})\right\|\right).$$

Hence, by Lemma C.1 and (22), one can easily claim that $\|\boldsymbol{A}^0\| \leq C/\sqrt{n}$ with probability at least $1 - e^{-cn}$ for some constants $c, C > 0$. On the other hand, since $\boldsymbol{v}(\boldsymbol{y} - f_t(\boldsymbol{X}))$ is rank-one and $\sigma'_\perp = \sigma' - \mu_1$ with $|\sigma'(x)| \leq \lambda_\sigma$, we can similarly obtain

$$\left\|\boldsymbol{B}^0\right\|_F \leq \frac{1}{n\sqrt{dh}}\left\|\boldsymbol{v}(\boldsymbol{y} - f_t(\boldsymbol{X})) \odot \sigma'_\perp(\boldsymbol{W}_t\boldsymbol{X}/\sqrt{d})\right\|_F \|\boldsymbol{X}\|$$
$$\leq \frac{1}{n\sqrt{hd}}\|\boldsymbol{X}\|(\|\boldsymbol{y}\| + \|f_0(\boldsymbol{X})\|)\|\boldsymbol{v}\|\max_{i,j}\left|\sigma'_\perp(\boldsymbol{W}_0\boldsymbol{X}/\sqrt{d})\right|_{i,j}$$
$$\leq \frac{\mu_1 + \lambda_\sigma}{\sqrt{n}}\frac{\|\boldsymbol{X}\|}{\sqrt{d}}\frac{\|\boldsymbol{v}\|}{\sqrt{h}}\frac{1}{\sqrt{n}}\left(\|\boldsymbol{y}\| + \frac{\|\boldsymbol{v}\|}{\sqrt{h}}\left\|\sigma(\boldsymbol{W}_0\boldsymbol{X}/\sqrt{d})\right\|\right).$$

As $\boldsymbol{A}^0$, we can apply Lemma C.1 and (22) again to conclude (20) for $t = 1$.

For general $t$, we apply induction. We assume that after the $t$-th gradient step with $\eta = \Theta(1)$, Eq. (20) holds for some constants $C, c > 0$. Following [6, Lemma 16], we now show that the similar high-probability statement also holds for $\boldsymbol{W}_{t+1}$ (for some different constants $c', C'$). Firstly,

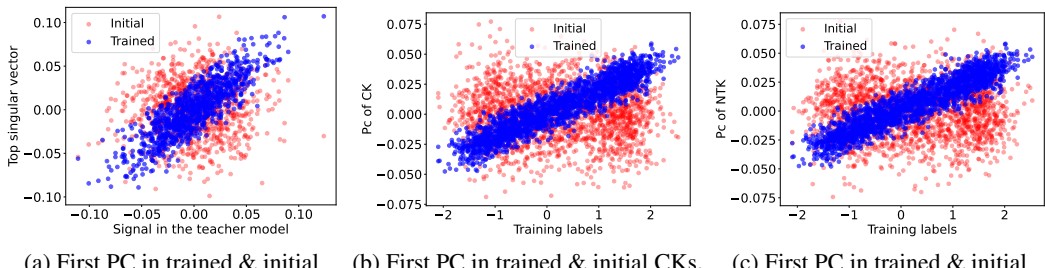

(a) First PC in trained & initial weights.

(b) First PC in trained & initial CKs.

(c) First PC in trained & initial NTKs.

Figure 27: Alignment between the leading PC of the weight/kernel matrices and the signal $\boldsymbol{\beta}$ or the training labels $\boldsymbol{y}$ before/after training for the experiment in Figure 26. We can observe strong alignments, in this case, comparing with Figure 25 because of outliers in above Figure 26. These kernel alignments induce anisotropic structures in the kernel matrices during training [78].

following the same argument as [71, Setion 6.6.1], we know that

$$\|f_t(\boldsymbol{X})\| \leq \|f_0(\boldsymbol{X})\| + \|f_t(\boldsymbol{X}) - f_0(\boldsymbol{X})\|$$

$$\leq \|f_0(\boldsymbol{X})\| + \frac{\lambda_\sigma}{\sqrt{h}}\|\boldsymbol{v}\|\frac{\|\boldsymbol{X}\|}{\sqrt{d}}\|\boldsymbol{W}_t - \boldsymbol{W}_0\|_F. \tag{23}$$

Note that $\|\boldsymbol{W}_t - \boldsymbol{W}_0\|_F = O(1/\sqrt{n})$ with high probability by the induction hypothesis. Hence, by Lemma C.1 and (22), we have $\|f_t(\boldsymbol{X})\| \leq C\sqrt{n}$ with high probability. Indeed, the difference between $f_t(\boldsymbol{X})$ and $f_0(\boldsymbol{X})$ is significantly negligible comparing with the initial value $f_0(\boldsymbol{X})$. Similarly with $\boldsymbol{A}_0$, $\boldsymbol{A}^t$ satisfies

$$\|\boldsymbol{A}^t\| = \|\boldsymbol{A}^t\|_F \leq \frac{\mu_1}{\sqrt{n}}\frac{\|\boldsymbol{X}\|}{\sqrt{d}}\frac{1}{\sqrt{n}}(\|\boldsymbol{y}\| + \|f_t(\boldsymbol{X})\|)\frac{\|\boldsymbol{v}\|}{\sqrt{h}}.$$

Analogously for $\boldsymbol{B}^t$, we have

$$\|\boldsymbol{B}^t\|_F \leq \frac{\mu_1 + \lambda_\sigma}{\sqrt{n}}\frac{\|\boldsymbol{X}\|}{\sqrt{d}}\frac{\|\boldsymbol{v}\|}{\sqrt{h}}\frac{1}{\sqrt{n}}(\|\boldsymbol{y}\| + \|f_t(\boldsymbol{X})\|).$$

Thus, Lemma C.1, (22), and (23) ensure that

$$\mathbb{P}\left(\|\boldsymbol{A}^t\|_F \geq \frac{C'}{\sqrt{n}}\right) \leq \exp\left(-c'n\right), \; \mathbb{P}\left(\|\boldsymbol{B}^t\|_F \geq \frac{C'}{\sqrt{n}}\right) \leq \exp\left(-c'n\right),$$

for constants $c', C' > 0$. Since $\|\boldsymbol{W}_{t+1} - \boldsymbol{W}_0\|_F \leq \|\boldsymbol{W}_t - \boldsymbol{W}_0\|_F + \eta\|\boldsymbol{A}^t\|_F + \eta\|\boldsymbol{B}^t\|_F$, by induction hypothesis, we can conclude that (20) holds for the $(t+1)$-th step with some constants $C, c > 0$, which are different from the constants at the $t$-th step. $\qquad\square$

As a corollary, by (17), we can also deduce the following norm bounds:

$$\mathbb{P}\left(\|\boldsymbol{W}_t - \boldsymbol{W}_0\| \geq \frac{C}{\sqrt{n}}\right) \leq \exp\left(-cn\right), \quad \mathbb{P}\left(\|\boldsymbol{W}_t - \boldsymbol{W}_0\|_{2,\infty} \geq \frac{C}{\sqrt{n}}\right) \leq \exp\left(-cn\right).$$

Lemma C.2 and the above bounds are empirically verified by Figure 28(a) for $t = 3$. Not only upper bounds, this simulation also shows that at early phase $\|\boldsymbol{W}_t - \boldsymbol{W}_0\|$, $\|\boldsymbol{W}_t - \boldsymbol{W}_0\|_F$, and $\|\boldsymbol{W}_t - \boldsymbol{W}_0\|_{2,\infty}$ are all of the same $\Theta(1/\sqrt{n})$ order.

As a remark, from the bound of the second term of (23), we can deduce that the change of the output of the NN satisfies

$$|f_t(\boldsymbol{x}) - f_0(\boldsymbol{x})| \leq \frac{C}{\sqrt{n}},$$

for some $t$-dependent constant $C > 0$, any $\boldsymbol{x} \sim \mathcal{N}(0, \mathbf{I})$ and any finite time $t$. In other words, when $\eta = \Theta(1)$, the change of the output of the NN at the early phase (i.e. $t = \Theta(1)$) is negligible and its order is $O(\frac{1}{\sqrt{n}})$.

## C.2 Proof of Lemma 4.1

In this section, we complete the proof of Lemma 4.1. We first mention the empirical validation of Lemma 4.1 in Figure 28. Notice that the changes in Frobenius norm for $\boldsymbol{W}$ and $\boldsymbol{K}^{\mathrm{CK}}$ are exactly $\Theta(1/\sqrt{n})$ and $\Theta(1/n)$, respectively. The operator norm of $\boldsymbol{K}^{\mathrm{NTK}}$ matches with Lemma 4.1, while the Frobenius norm of the change decays slower than the rate $\Theta(1/n)$. Additionally, in the simulation, we use $\boldsymbol{v} \sim \mathcal{N}(0, \mathbf{I})$, which indicates that our assumption for $\boldsymbol{v}$ in Lemma 4.1 can be weakened.

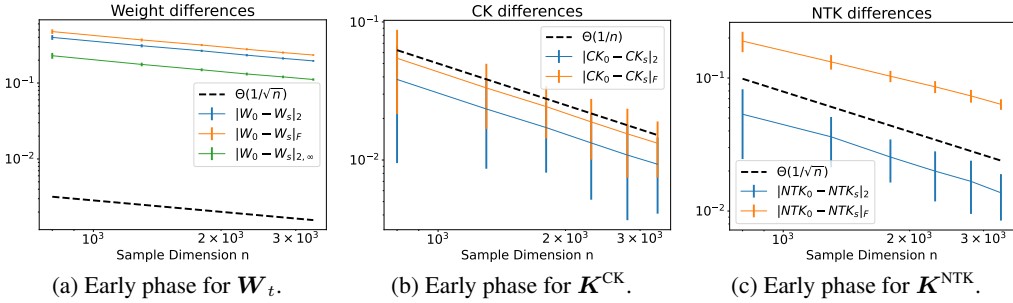

(a) Early phase for $\boldsymbol{W}_t$.     (b) Early phase for $\boldsymbol{K}^{\mathrm{CK}}$.     (c) Early phase for $\boldsymbol{K}^{\mathrm{NTK}}$.

Figure 28: Empirical validations for Lemma 4.1 and Lemma C.2 at $t = 3$. Here $\sigma_\varepsilon = 0.2$, activation $\sigma$ is a normalized ReLU and the target function $\sigma^*$ is normalized $\tanh$. Fix $d/n = 0.6$ and $N/n = 1.2$ as $n$ is increasing. At each dimension, we take 25 trials to average. (a) Norms of the changes for $\boldsymbol{W}_3 - \boldsymbol{W}_0$. (b) Norms of the changes for $\boldsymbol{K}_3^{\mathrm{CK}} - \boldsymbol{K}_0^{\mathrm{CK}}$. (c) Norms of the changes for $\boldsymbol{K}_3^{\mathrm{NTK}} - \boldsymbol{K}_0^{\mathrm{NTK}}$.

*Proof of Lemma 4.1.* Lemma C.2 directly validates the control of $\frac{1}{\sqrt{d}} \|\boldsymbol{W}_t - \boldsymbol{W}_0\|_F$. By virtue of this result, we now present estimates for CK and NTK. Based on [71, Section 6.6.1], we have

$$\left\| \sigma(\boldsymbol{W}_0 \boldsymbol{X}/\sqrt{d}) - \sigma(\boldsymbol{W}_t \boldsymbol{X}/\sqrt{d}) \right\| \leq \frac{\lambda_\sigma}{\sqrt{d}} \|\boldsymbol{X}\| \|\boldsymbol{W}_0 - \boldsymbol{W}_t\|_F. \tag{24}$$

We apply the mean value theorem to obtain this inequality. Recall the operator norm bound for Gaussian random matrix $\boldsymbol{X}$ in (22). We know $\|\boldsymbol{X}/\sqrt{d}\| \lesssim 1 + \sqrt{\gamma_1}$ with high probability as $n/d \to \gamma_1$. Hence, with the help of Lemma C.2, we can claim

$$\left\| \sigma(\boldsymbol{W}_0 \boldsymbol{X}/\sqrt{d}) - \sigma(\boldsymbol{W}_t \boldsymbol{X}/\sqrt{d}) \right\| \leq C \lambda_\sigma (1 + \sqrt{\gamma_1})/\sqrt{n},$$

with probability at least $1 - \exp(-cn)$, for any fixed finite $t \in [n]$. Similarly, we can control the change in the Frobenius norm as follows:

$$\left\| \sigma(\boldsymbol{W}_0 \boldsymbol{X}/\sqrt{d}) - \sigma(\boldsymbol{W}_t \boldsymbol{X}/\sqrt{d}) \right\|_F^2 \leq \frac{\lambda_\sigma^2}{d} \|\boldsymbol{X}\|^2 \|\boldsymbol{W}_0 - \boldsymbol{W}_t\|_F^2 \leq C \lambda_\sigma^2 (1 + \sqrt{\gamma_1})^2/n, \tag{25}$$

with probability at least $1 - \exp(-cn)$. Therefore, we can control the change in the CK matrix in the Frobenius norm by the following inequalities:

$$\left\| \boldsymbol{K}_t^{\mathrm{CK}} - \boldsymbol{K}_0^{CK} \right\|_F$$

$$\leq \frac{1}{h} \left\| \sigma(\boldsymbol{W}_t \boldsymbol{X}/\sqrt{d})^\top \left( \sigma(\boldsymbol{W}_t \boldsymbol{X}/\sqrt{d}) - \sigma(\boldsymbol{W}_0 \boldsymbol{X}/\sqrt{d}) \right) \right\|_F + \frac{1}{h} \left\| \sigma(\boldsymbol{W}_0 \boldsymbol{X}/\sqrt{d})^\top \left( \sigma(\boldsymbol{W}_t \boldsymbol{X}/\sqrt{d}) - \sigma(\boldsymbol{W}_0 \boldsymbol{X}/\sqrt{d}) \right) \right\|_F$$

$$\leq \frac{1}{h} \left( \left\| \sigma(\boldsymbol{W}_t \boldsymbol{X}/\sqrt{d}) - \sigma(\boldsymbol{W}_0 \boldsymbol{X}/\sqrt{d}) \right\| + \left\| \sigma(\boldsymbol{W}_0 \boldsymbol{X}/\sqrt{d}) \right\| \right) \cdot \left\| \sigma(\boldsymbol{W}_t \boldsymbol{X}/\sqrt{d}) - \sigma(\boldsymbol{W}_0 \boldsymbol{X}/\sqrt{d}) \right\|_F$$

$$+ \frac{1}{h} \left\| \sigma(\boldsymbol{W}_0 \boldsymbol{X}/\sqrt{d}) \right\| \cdot \left\| \sigma(\boldsymbol{W}_t \boldsymbol{X}/\sqrt{d}) - \sigma(\boldsymbol{W}_0 \boldsymbol{X}/\sqrt{d}) \right\|_F.$$

Therefore, by (24), (25) and Lemma C.1, we can claim that there exist constants $c, C > 0$ such that with probability at least $1 - \exp(-cn)$, $\left\| \boldsymbol{K}_t^{\mathrm{CK}} - \boldsymbol{K}_0^{CK} \right\|_F$ is upper bounded by $C/n$ in the LWR.

Now we consider the change in the NTK matrix during training. Since the empirical NTK can be decomposed into two parts, one of which is exactly the CK, it suffices to consider the change of the first part of the empirical NTK. Recall that

$$\boldsymbol{K}_t := \frac{1}{d} \boldsymbol{X}^\top \boldsymbol{X} \odot \frac{1}{h} \sigma'\left( \frac{1}{\sqrt{d}} \boldsymbol{W}_t \boldsymbol{X} \right)^\top \mathrm{diag}(\boldsymbol{v}_t)^2 \sigma'\left( \frac{1}{\sqrt{d}} \boldsymbol{W}_t \boldsymbol{X} \right).$$

Following the notation in [71], we denote $\mathcal{J}(\boldsymbol{W}_t) := [\mathcal{J}(\boldsymbol{w}_1^t), \ldots, \mathcal{J}(\boldsymbol{w}_N^t)] \in \mathbb{R}^{n \times hd}$ with $\mathcal{J}(\boldsymbol{w}_i) := \frac{v_i}{\sqrt{h}} \operatorname{diag}(\sigma'(\boldsymbol{X}^\top \boldsymbol{w}_i/\sqrt{d})) \boldsymbol{X}^\top/\sqrt{d} \in \mathbb{R}^{n \times d}$. Hence, $\boldsymbol{K}_t = \mathcal{J}(\boldsymbol{W}_t)\mathcal{J}(\boldsymbol{W}_t)^\top$ and

$$
\begin{aligned}
\|\boldsymbol{K}_t - \boldsymbol{K}_0\| &= \left\| \mathcal{J}(\boldsymbol{W}_t)\mathcal{J}(\boldsymbol{W}_t)^\top - \mathcal{J}(\boldsymbol{W}_0)\mathcal{J}(\boldsymbol{W}_0)^\top \right\| \\
&\leq 2 \|\mathcal{J}(\boldsymbol{W}_0)\| \|\mathcal{J}(\boldsymbol{W}_t) - \mathcal{J}(\boldsymbol{W}_0)\| + \|\mathcal{J}(\boldsymbol{W}_t) - \mathcal{J}(\boldsymbol{W}_0)\|^2. 
\end{aligned} \tag{26}
$$

By [71, Lemma 6.6], we know $\|\mathcal{J}(\boldsymbol{W}_0)\|^2 = \|\boldsymbol{K}_0^{\mathrm{NTK}}\|$ is upper bounded by some constant $C > 0$ with high probability. Then, we apply the inequalities from Lemma 6.5 of [71] to obtain

$$
\begin{aligned}
&\|\mathcal{J}(\boldsymbol{W}_t) - \mathcal{J}(\boldsymbol{W}_0)\|^2 \\
&= \left\| \left( \left(\sigma'(\boldsymbol{W}_t \boldsymbol{X}/\sqrt{d}) - \sigma'(\boldsymbol{W}_0 \boldsymbol{X}/\sqrt{d})\right)^\top \frac{\operatorname{diag}(\boldsymbol{v})^2}{h} \left(\sigma'(\boldsymbol{W}_t \boldsymbol{X}/\sqrt{d}) - \sigma'(\boldsymbol{W}_0 \boldsymbol{X}/\sqrt{d})\right) \right) \odot \left( \frac{\boldsymbol{X}^\top \boldsymbol{X}}{d} \right) \right\| \\
&\leq \left\| \left(\sigma'(\boldsymbol{W}_t \boldsymbol{X}/\sqrt{d}) - \sigma'(\boldsymbol{W}_0 \boldsymbol{X}/\sqrt{d})\right)^\top \frac{\operatorname{diag}(\boldsymbol{v})}{\sqrt{h}} \right\|^2 \left( \max_{i \in [n]} \left\| \boldsymbol{x}_i/\sqrt{d} \right\|^2 \right) \\
&\leq \frac{1}{h} \|\boldsymbol{v}\|_\infty^2 \left\| \sigma'(\boldsymbol{W}_t \boldsymbol{X}/\sqrt{d}) - \sigma'(\boldsymbol{W}_0 \boldsymbol{X}/\sqrt{d}) \right\|^2 \left( \max_{i \in [n]} \left\| \boldsymbol{x}_i/\sqrt{d} \right\|^2 \right) \\
&\leq \frac{\lambda_\sigma^2}{h} \frac{\|\boldsymbol{X}\|^2}{d} \|\boldsymbol{W}_t - \boldsymbol{W}_0\|_F^2 \left( \max_{i \in [n]} \left\| \boldsymbol{x}_i/\sqrt{d} \right\|^2 \right), 
\end{aligned} \tag{27}
$$

where the last inequality is due to the mean value theorem, the uniform bound on $\sigma''$, and the assumption on the second layer $\boldsymbol{v}$. Notice that Gaussian random vectors satisfy

$$
\mathbb{P}\left( \max_{i \in [n]} \frac{1}{d} \|\boldsymbol{x}_i\|^2 \geq 2 \right) \leq 2ne^{-cn}, \tag{28}
$$

as $n/d \to \gamma_1$ and $h/d \to \gamma_2$. Thus, with (22) and Lemma C.2, we obtain

$$
\mathbb{P}\left( \|\mathcal{J}(\boldsymbol{W}_t) - \mathcal{J}(\boldsymbol{W}_0)\| \geq \frac{C\lambda_\sigma(1 + \gamma_1)}{n} \right) \leq 4ne^{-cn},
$$

where constant $C$ relies on the number of steps $t$. Hence, by (26), we can finally bound in norm the difference between the initial and the trained NTK matrices at the early phase ($t$ is finite).

$\square$

**Corollary C.3.** *For any fixed $t \in \mathbb{N}$, $i \in [d]$ and $k \in [n]$, denote $\lambda_i^t$, $\nu_k^t$ and $\mu_k^t$ the $i$-th, and $k$-th eigenvalues of $\frac{1}{h}\boldsymbol{W}_t^\top \boldsymbol{W}_t$, $\boldsymbol{K}_t^{CK}$ and $\boldsymbol{K}_t^{NTK}$, respectively. Then, under the assumptions of Lemma 4.1, we have*

$$
|\lambda_i^t - \lambda_i^0|, \ |\nu_k^t - \nu_k^0|, \ |\mu_k^t - \mu_k^0| \to 0,
$$

*almost surely in LWR. Consequently, the eigenvalues of $\frac{1}{h}\boldsymbol{W}_t^\top \boldsymbol{W}_t$, $\boldsymbol{K}_t^{CK}$ and $\boldsymbol{K}_t^{NTK}$ are the same as corresponding the eigenvalues of initial $\frac{1}{h}\boldsymbol{W}_0^\top \boldsymbol{W}_0$, $\boldsymbol{K}_0^{CK}$ and $\boldsymbol{K}_0^{NTK}$, respectively.*

This corollary is a direct outcome of Weyl's inequality from Theorem A.46 in [8]. Consequently, this corollary concludes that for any fixed $t \geq 0$, almost surely, the limiting spectra of $\frac{1}{h}\boldsymbol{W}_t^\top \boldsymbol{W}_t$, $\boldsymbol{K}_t^{CK}$ and $\boldsymbol{K}_t^{NTK}$ are the same as those of $\frac{1}{h}\boldsymbol{W}_0^\top \boldsymbol{W}_0$, $\boldsymbol{K}_0^{CK}$ and $\boldsymbol{K}_0^{NTK}$ in LWR. This corollary claims that not only does the bulk of distributions stay identical to the initialization, but also that any eigenvalues stay the same as at the initialization. This shows that the smallest eigenvalue of $\boldsymbol{K}_t^{NTK}$ has the same lower bound as $\boldsymbol{K}_0^{NTK}$ in the early phase of training.

### C.3 Global Convergence for GD Under LWR

In this section, we study the final stage of (10) as training loss is approaching zero and prove Theorem 4.2. Figure 29 shows that the spectra are unchanged globally, even after training in this case. In Corollary 4.3, we confirm this observation for the weight, CK, and NTK matrices via Frobenius norm control. In the simulation, the second layer is initialized as $\boldsymbol{v} \sim \mathcal{N}(0, \mathbf{I})$, which is more general than our assumption on $\boldsymbol{v}$ in Theorem 4.2.

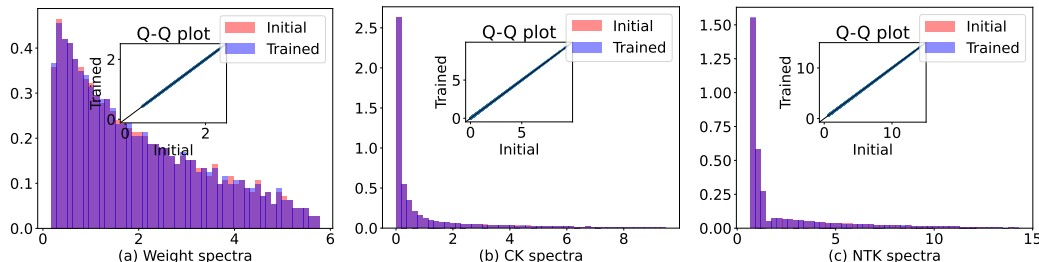

Figure 29: The initial and trained spectra (until training loss is less than $10^{-5}$) when using GD only for the first layer ($h = 3000, n = 2000, d = 1000$): (a) Weight spectra. (b) $\boldsymbol{K}^{\text{CK}}$ spectra. (c) $\boldsymbol{K}^{\text{NTK}}$ spectra. The final $R^2$ score is 0.55964 and the test loss is 0.44724. The activation is a normalized ReLU, and the target is Sigmoid.

*Proof of Theorem 4.2.* Recall the Jacobian matrix $\mathcal{J}(\boldsymbol{W})$ defined in the proof of Lemma 4.1, and the definition of $\alpha$ based on (11) in Section 3. Denote the event

$$\mathcal{A} := \left\{ \|\boldsymbol{X}\| \leq 2\left(1 + \sqrt{\gamma_1}\right)\sqrt{d}, \ \max_{i \in [n]} \|\boldsymbol{x}_i\|^2 \leq 2d, \sigma_{\min}(\mathcal{J}(\boldsymbol{W}_0)) \geq 2\alpha \right\}.$$

By (22), (28) and Theorem 2.9 of [81], we have $\mathbb{P}\left(\mathcal{A}\right) \geq 1 - 2e^{-cn} - 2ne^{-cn} - n^{-7/3}$ for some constant $c > 0$ and all large $n$ in LWR. In the following, conditionally on event $\mathcal{A}$, we will apply Theorem 6.10 of [71] to obtain the global convergence. Conditionally on $\mathcal{A}$, Lemma 6.6 of [71] implies

$$\|\mathcal{J}(\boldsymbol{W})\| \leq \lambda_\sigma \|\boldsymbol{v}\|_\infty \left\|\boldsymbol{X}/\sqrt{d}\right\| \leq 2\lambda_\sigma(1 + \sqrt{\gamma_1}), \tag{29}$$

for any $\boldsymbol{W}$. Define $\beta = 2\lambda_\sigma(1 + \sqrt{\gamma_1})$. Moreover, in terms of (27), we can verify the Lipschitz property for the Jacobian matrix as follows: conditionally on $\mathcal{A}$,

$$\left\|\mathcal{J}(\tilde{\boldsymbol{W}}) - \mathcal{J}(\boldsymbol{W})\right\| \leq \frac{2\beta}{\sqrt{h}} \left\|\tilde{\boldsymbol{W}} - \boldsymbol{W}\right\|_F, \tag{30}$$

for any $\tilde{\boldsymbol{W}}, \boldsymbol{W} \in \mathbb{R}^{h \times d}$. Therefore, conditionally on $\mathcal{A}$, $\mathcal{J}(\boldsymbol{W})$ is a $L$-Lipschitz function with respect to $\boldsymbol{W}$ where $L := \frac{2\beta}{\sqrt{h}}$. To complete the proof, it suffices to investigate the smallest singular value of $\mathcal{J}(\boldsymbol{W})$ when $\boldsymbol{W}$ is in the vicinity of $\boldsymbol{W}_0$. Recall $\ell(\boldsymbol{W}) = \|\boldsymbol{y} - f_{\boldsymbol{W}}(\boldsymbol{X})\|$. Notice that for any unit vector $\boldsymbol{u} \in \mathbb{R}^n$, we have $\boldsymbol{u}^\top f_{\boldsymbol{W}_0}(\boldsymbol{X}) = \frac{1}{\sqrt{h}} \sum_{i=1}^h v_i \sigma(\boldsymbol{w}_i^\top \boldsymbol{X}/\sqrt{d})\boldsymbol{u}$, where $\boldsymbol{w}_i^\top$ is the $i$-th row of $\boldsymbol{W}_0$ for $i \in [N]$. Consider event $\mathcal{B} := \left\{ \left\|\sigma(\boldsymbol{W}_0 \boldsymbol{X}/\sqrt{d})\right\| \leq C\sqrt{n} \right\}$ for some universal constant $C > 0$. Lemma C.1 proves $\mathbb{P}(\mathcal{B}) \geq 1 - 2e^{-cn}$. By the assumption of $\boldsymbol{v}$, we know each entry $v_i$ is a sub-Gaussian random variable with a sub-Gaussian norm at most 1. Then, according to Hoeffding's inequality, conditionally on the event $\mathcal{B}$, we have

$$\mathbb{P}\left( \left| \frac{1}{\sqrt{h}} \sum_{i=1}^h v_i \sigma(\boldsymbol{w}_i^\top \boldsymbol{X}/\sqrt{d})\boldsymbol{u} \right| \geq t \right) \leq 2\exp\left(-ct^2\right),$$

for every $t \geq 0$ and some constant $c > 0$. Let $t = 2\sqrt{n}$. Considering an $\frac{1}{4}$-net $\mathcal{N}$ of the unit sphere $\mathbb{S}^{n-1}$, we can get

$$\mathbb{P}\left(\|f_{\boldsymbol{W}_0}(\boldsymbol{X})\| \geq \sqrt{n}\right) \leq \mathbb{P}\left(2\max_{\boldsymbol{u} \in \mathcal{N}} \left|\boldsymbol{u}^\top f_{\boldsymbol{W}_0}(\boldsymbol{X})\right| \geq \sqrt{n}\right) \leq 9^n 2\exp\left(-cn\right) \leq 2e^{-c'n}, \tag{31}$$

for some constant $c' > 0$. Hence, based on Lemma C.1 and (31), we can obtain $\ell(\boldsymbol{W}_0) \leq C_0\sqrt{n}$ with high probability for some universal constant $C_0 > 0$. Let us denote this event as $\mathcal{C} : \{\ell(\boldsymbol{W}_0) \leq C_0\sqrt{n}\}$. Define $R := 4\ell(\boldsymbol{W}_0)/\alpha$. For any $\boldsymbol{W}$ in a ball of radius $R$ centered at $\boldsymbol{W}_0$, we have $\|\boldsymbol{W}_0 - \boldsymbol{W}\|_F \leq R$ and $\|\mathcal{J}(\boldsymbol{W}) - \mathcal{J}(\boldsymbol{W}_0)\| \leq LR$, conditionally on event $\mathcal{A}$. Thus, by (30), on event $\mathcal{A} \cap \mathcal{C}$, the smallest singular value $\sigma_{\min}(\mathcal{J}(\boldsymbol{W}))$ of the Jacobian matrix $\mathcal{J}(\boldsymbol{W})$ can be bounded by

$$\sigma_{\min}(\mathcal{J}(\boldsymbol{W})) \geq \ \sigma_{\min}(\mathcal{J}(\boldsymbol{W}_0)) - \|\mathcal{J}(\boldsymbol{W}) - \mathcal{J}(\boldsymbol{W}_0)\|$$

$$\geq \ 2\alpha - LR \geq 2\alpha - \frac{8\beta}{\alpha}\frac{\ell(\boldsymbol{W}_0)}{\sqrt{h}} \geq 2\alpha - \frac{8C\beta}{\alpha}\sqrt{\frac{\gamma_1}{\gamma_2}},$$

for some universal constant $C > 0$ and sufficiently large $n, d, h$. Notice that here constants $C, \beta$, and $\alpha$ do not rely on $\gamma_2$. Therefore, there exists a sufficiently large $\gamma^* > 0$ such that for all $\gamma_2 \geq \gamma^*$, we have $2\alpha - \frac{8C\beta}{\alpha}\sqrt{\frac{\gamma_1}{\gamma_2}} \geq \alpha$. In other words, when $h$ is sufficiently large but still in the same order as $n$ and $d$, for all $\|\boldsymbol{W} - \boldsymbol{W}_0\|_F \leq R$, we have $\sigma_{\min}(\mathcal{J}(\boldsymbol{W})) \geq \alpha$ conditionally on $\mathcal{C} \cap \mathcal{A}$. Combining with (29) and (30), conditionally on $\mathcal{C} \cap \mathcal{A}$, all the assumptions of Theorem 6.10 by [71] are satisfied when $\|\boldsymbol{W} - \boldsymbol{W}_0\|_F \leq R$. Therefore, when the learning rate $\frac{\eta}{n} \leq \frac{1}{\beta^2}\min\left\{1, \frac{4\alpha}{LR}\right\}$, we can get (12)-(14) for all $t \in \mathbb{N}$, conditionally on $\mathcal{C} \cap \mathcal{A}$. Both events $\mathcal{A}$ and $\mathcal{C}$ occur with high probability and only depend on initialization $\boldsymbol{W}_0$, $\boldsymbol{X}$ and $\boldsymbol{y}$. Hence we complete the proof of this theorem. Notice that since $\gamma_2 \geq \gamma^*$ is sufficiently large, $\frac{4\alpha}{LR} \geq \frac{\alpha^2}{2C\beta}\sqrt{\frac{\gamma_2}{\gamma_1}} > 1$. Therefore, it suffices to require $\eta \leq n/\beta^2$ to conclude that (12), (13) and (14) hold with high probability. This completes the proof. Moreover, (14) further shows that for all $t \in \mathbb{N}$,

$$\|\boldsymbol{W}_0 - \boldsymbol{W}_t\|_F \leq R \leq C\sqrt{n} + o_{d,\mathbb{P}}(1), \tag{32}$$

where we again apply Lemma C.1 in the following way:

$$\ell(\boldsymbol{W}_0) \leq C\sqrt{n} + o_{d,\mathbb{P}}(1),$$

for some constant $C > 0$ only depending on $\gamma_1, \gamma_2, \sigma_\varepsilon, \sigma$ and $\sigma^*$. $\qquad \square$

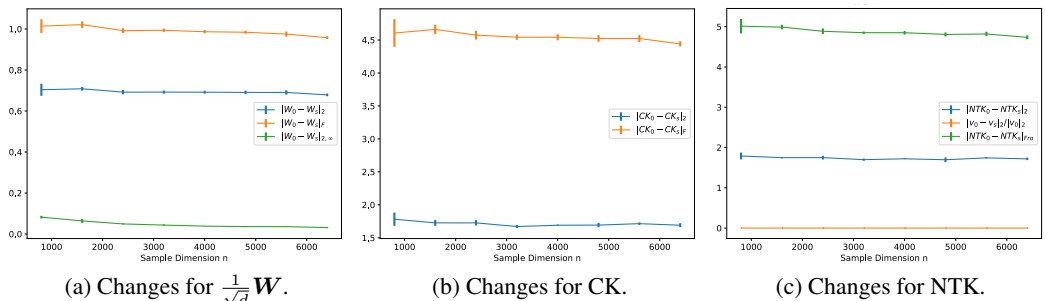

(a) Changes for $\frac{1}{\sqrt{d}}\boldsymbol{W}$.      (b) Changes for CK.      (c) Changes for NTK.

Figure 30: Measuring the change for the weight, CK, and NTK matrices when training NN with (10). We fix $d/n = 1.2$ and $h/n = 0.6$ when $n$ is increasing. Here, $\sigma$ is normalized ReLU and the target is normalized $\tanh$. The largest $n = 6400$ and the learning rate $\eta = 5.0$ for all training processes. We train each neural network until the training losses approach zero. Each experiment repeats 4 times. In (a), we consider the changes $\frac{1}{\sqrt{d}}\|\boldsymbol{W}_t - \boldsymbol{W}_0\|, \frac{1}{\sqrt{d}}\|\boldsymbol{W}_t - \boldsymbol{W}_0\|_F$, and $\frac{1}{\sqrt{d}}\|\boldsymbol{W}_t - \boldsymbol{W}_0\|_{2,\infty}$.

As a corollary, (14) controls the deviation of the final step weight from the initial weight. This is empirically shown in Figure 30(a), which shows that $\frac{1}{\sqrt{d}}\|\boldsymbol{W}_t - \boldsymbol{W}_0\|, \frac{1}{\sqrt{d}}\|\boldsymbol{W}_t - \boldsymbol{W}_0\|_F$, and $\frac{1}{\sqrt{d}}\|\boldsymbol{W}_t - \boldsymbol{W}_0\|_{2,\infty}$ are $\Theta(1)$ when trainable parameters are convergent. This implies that the final $\boldsymbol{W}_t$ is still close to the initial weight $\boldsymbol{W}_0$, even after training. Consequently, with this observation, we can prove Corollary 4.3 in the following.

*Proof of Corollary 4.3.* Based on (32), we know $\frac{1}{\sqrt{d}}\|\boldsymbol{W}_o - \boldsymbol{W}_t\|_F \leq C_0$ holds with high probability for some universal constant $C_0 > 0$. Conditionally on this event, we can then estimate changes in CK and NTK after training. The method is analogous to Lemma 4.1. For CK, we employ Lemma C.1 and (25) to get

$$\left\|\boldsymbol{K}_t^{\mathrm{CK}} - \boldsymbol{K}_0^{CK}\right\|_F$$
$$\leq \frac{2}{h}\left(\left\|\sigma(\boldsymbol{W}_t\boldsymbol{X}/\sqrt{d}) - \sigma(\boldsymbol{W}_0\boldsymbol{X}/\sqrt{d})\right\| + \left\|\sigma(\boldsymbol{W}_0\boldsymbol{X}/\sqrt{d})\right\|\right) \cdot \left\|\sigma(\boldsymbol{W}_t\boldsymbol{X}/\sqrt{d}) - \sigma(\boldsymbol{W}_0\boldsymbol{X}/\sqrt{d})\right\|_F$$
$$\lesssim \frac{2\lambda_\sigma^2(1 + \sqrt{\gamma_1})^2}{h}\|\boldsymbol{W}_0 - \boldsymbol{W}_t\|_F^2 + \frac{2C\sqrt{n}\lambda_\sigma(1 + \sqrt{\gamma_1})}{h}\|\boldsymbol{W}_0 - \boldsymbol{W}_t\|_F$$
$$\lesssim \frac{2\lambda_\sigma(1 + \sqrt{\gamma_1})C_0}{\gamma_2^2}\left(\lambda_\sigma(1 + \sqrt{\gamma_1})C_0 + C\sqrt{\gamma_1}\right) = O_{d,\mathbb{P}}(1).$$

Hence, this shows control of the change for the CK matrix after training, compared with the initial CK.

Let us denote $\boldsymbol{w}_i^t \in \mathbb{R}^{1 \times d}$ as the $i$-th row of $\boldsymbol{W}_t$, and $\boldsymbol{x}_j$ as the $j$-th column of $\boldsymbol{X}$. Additionally, by Assumption 3.2, we know that

$$|\sigma'(x) - \sigma'(y)| \le \lambda_\sigma |x - y|, \tag{33}$$

for any $x, y \in \mathbb{R}$. For NTK, by modifying (27), one can deduce that

$$\|\mathcal{J}(\boldsymbol{W}_t) - \mathcal{J}(\boldsymbol{W}_0)\|_F^2 = \sum_{i=1}^{h} \left\|\mathcal{J}(\boldsymbol{w}_i^t) - \mathcal{J}(\boldsymbol{w}_i^0)\right\|_F^2$$

$$\overset{(i)}{\le} \frac{1}{h} \sum_{i=1}^{h} \left\|\operatorname{diag}\left(\sigma'(\boldsymbol{w}_i^t \boldsymbol{X}/\sqrt{d} - \sigma'(\boldsymbol{w}_i^0 \boldsymbol{X}/\sqrt{d})\right)\right\|_F^2 \left\|\frac{\boldsymbol{X}}{\sqrt{d}}\right\|^2$$

$$\overset{(ii)}{\le} \frac{(1+\sqrt{\gamma_1})^2}{h} \sum_{i=1}^{h} \left\|\operatorname{diag}\left(\sigma'(\boldsymbol{w}_i^t \boldsymbol{X}/\sqrt{d} - \sigma'(\boldsymbol{w}_i^0 \boldsymbol{X}/\sqrt{d})\right)\right\|_F^2 + o_{d,\mathbb{P}}(1)$$

$$\overset{(iii)}{\le} \frac{\lambda_\sigma^2 (1+\sqrt{\gamma_1})^2}{h} \sum_{i=1}^{h} \sum_{j=1}^{n} \left(\frac{1}{\sqrt{d}}(\boldsymbol{w}_i^t - \boldsymbol{w}_i^0)\boldsymbol{x}_j\right) + o_{d,\mathbb{P}}(1)$$

$$\overset{(iv)}{\le} \frac{\lambda_\sigma^2 (1+\sqrt{\gamma_1})^4}{h} \|\boldsymbol{W}_t - \boldsymbol{W}_0\|_F^2 + o_{d,\mathbb{P}}(1) \le \frac{\lambda_\sigma^2 (1+\sqrt{\gamma_1})^4 C_0^2}{\gamma_2} + o_{d,\mathbb{P}}(1),$$

where $(i)$ is because of [80, Exercise 6.3.3] and the assumption on $\boldsymbol{v}$, $(ii)$ is due to (22), $(iii)$ is due to the definition of Frobenius norm and (33), and $(iv)$ is due to [80, Exercise 6.3.3] and (22). As a result, from (26), we can finally conclude that $\left\|\boldsymbol{K}_t^{\text{CK}} - \boldsymbol{K}_0^{CK}\right\|_F = O_{d,\mathbb{P}}(1)$ as $n/d \to \gamma_1$ and $h/d \to \gamma_2$.

As for the limiting spectra of weight and kernel matrices, since we know that

$$\frac{1}{\sqrt{d}} \|\boldsymbol{W}_t - \boldsymbol{W}_0\|_F, \ \left\|\boldsymbol{K}_t^{\text{CK}} - \boldsymbol{K}_0^{\text{CK}}\right\|_F, \ \left\|\boldsymbol{K}_t^{\text{NTK}} - \boldsymbol{K}_0^{\text{NTK}}\right\|_F = O_{d,\mathbb{P}}(1),$$

we can automatically apply Corollary A.41 of [8]. This directly implies that the limiting empirical spectra of $\frac{1}{h}\boldsymbol{W}_t^\top \boldsymbol{W}_t$, $\boldsymbol{K}_t^{\text{CK}}$ and $\boldsymbol{K}_t^{\text{NTK}}$ are the same as the limiting spectra of $\frac{1}{h}\boldsymbol{W}_0^\top \boldsymbol{W}_0$, $\boldsymbol{K}_0^{\text{CK}}$ and $\boldsymbol{K}_0^{\text{NTK}}$, respectively, as $n/d \to \gamma_1$ and $h/d \to \gamma_2$ (see Figure 29). $\qquad \square$

**Further Simulations for Changes in Norms.** The simulation of Figure 30 empirically coincides with the norm bounds in Theorem 4.2 for different norms. Because of (17), it suffices to only consider the Frobenius norm of the change for each matrix. As a remark, Theorem 4.2 requires $\gamma_2$ to be larger than some threshold $\gamma^*$ to ensure norms of the change throughout training. However, Figure 30 indicates Theorem 4.2 still holds even when $\gamma^*$ is small i.e. the width $h$ is not very large. Here in Figure 30, $\gamma_2 = \frac{1}{2}\gamma_1 = 0.6$. Figure 31 suggests that similar results to Theorem 4.2 and Corollary 4.3 hold for SGD when training the first layer. This is also akin to Figure 21. Moreover, we further conjecture that similar results to Theorem 4.2 and Corollary 4.3 will hold even when training both layers (3). Denote the second layer at step $t$ by $\boldsymbol{v}_t$. Then, indicated by Figure 32, $\frac{1}{\sqrt{d}} \|\boldsymbol{W}_t - \boldsymbol{W}_0\|, \frac{1}{\sqrt{d}} \|\boldsymbol{W}_t - \boldsymbol{W}_0\|_F, \frac{1}{\sqrt{d}} \|\boldsymbol{W}_t - \boldsymbol{W}_0\|_{2,\infty}$, $\left\|\boldsymbol{K}_t^{\text{CK}} - \boldsymbol{K}_0^{\text{CK}}\right\|, \left\|\boldsymbol{K}_t^{\text{CK}} - \boldsymbol{K}_0^{\text{CK}}\right\|_F, \left\|\boldsymbol{K}_t^{\text{NTK}} - \boldsymbol{K}_0^{\text{NTK}}\right\|$, and $\|\boldsymbol{v}_t - \boldsymbol{v}_0\| / \|\boldsymbol{v}_0\|$ are all $\Theta(1)$ in LWR. Since $\|\boldsymbol{v}_0\| = O(\sqrt{h})$, we observe that $\|\boldsymbol{v}_t - \boldsymbol{v}_0\| = \Theta(\sqrt{h})$ in this case. Meanwhile, unlike Corollary 4.3, Figure 32(c) cannot verify that $\left\|\boldsymbol{K}_t^{\text{NTK}} - \boldsymbol{K}_0^{\text{NTK}}\right\|_F$ is still upper bounded by a constant. One possible explanation is that in this case the change in the second layer $\boldsymbol{v}_t$ is much more significant than in the first-layer weight $\boldsymbol{W}_t$, hence the NTK matrix may change a lot in this general case.

Similarly, Figure 33 shows norms of the change when training the NN with SGD for both layers. Here we use the same batch size 128 and learning rate $\eta = 1.0$ for all experiments when varying the dimension $n$ but fixing the aspect ratios $\gamma_1$ and $\gamma_2$. All the observations are similar to the results of Corollary 4.3 except the Frobenius norm of the change for NTK. Figure 33(c) indicates that $\left\|\boldsymbol{K}_t^{\text{NTK}} - \boldsymbol{K}_0^{\text{NTK}}\right\|_F$ will not be $\Theta(1)$ anymore, which fluctuates more in this case.

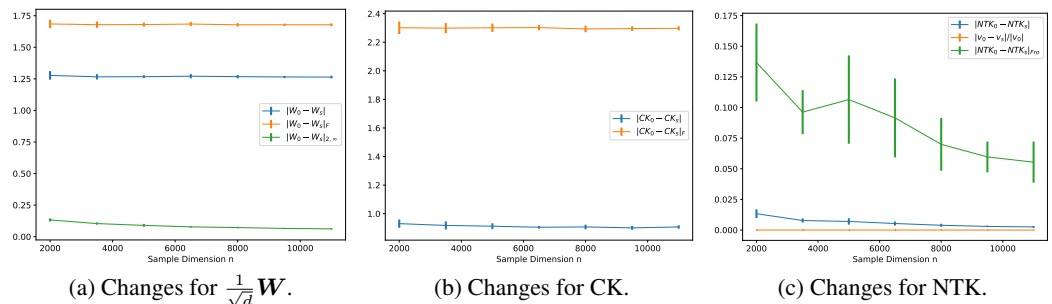

(a) Changes for $\frac{1}{\sqrt{d}}\boldsymbol{W}$.     (b) Changes for CK.     (c) Changes for NTK.

Figure 31: Measuring the change for weight $\boldsymbol{W}_t$, $\boldsymbol{K}^{\text{CK}}$, and $\boldsymbol{K}^{\text{NTK}}$ matrices when training NN with (10) for the first layer $\boldsymbol{W}_t$ using SGD with batch size 200. We fix $d/n = 0.6$ and $h/n = 1.2$ when $n$ is increasing. Here $\sigma$ is normalized softplus and the target is normalized $\tanh$. The largest $n$ is 11000, $\sigma_\varepsilon = 0.3$ and the learning rate is $\eta = 3.6$ for all training processes. We train each neural network until the training losses approach zero. Each experiment repeats 15 times.

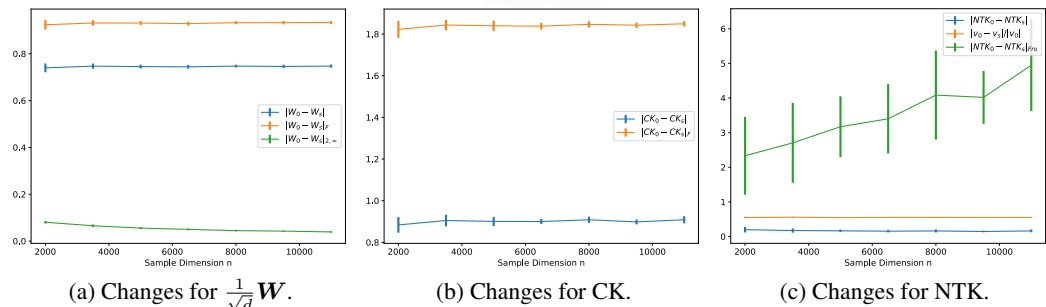

(a) Changes for $\frac{1}{\sqrt{d}}\boldsymbol{W}$.     (b) Changes for CK.     (c) Changes for NTK.

Figure 32: Measuring the change for $\boldsymbol{W}_t$, $\boldsymbol{K}^{\text{CK}}$, the second layer $\boldsymbol{v}_t$ and $\boldsymbol{K}^{\text{NTK}}$ when training NN with (3) for both layer $\boldsymbol{W}_t$ and $\boldsymbol{v}_t$ using GD. We fix $d/n = 0.5$ and $h/n = 0.8$ when $n$ is increasing. Here, $\sigma$ is normalized softplus and the target is normalized $\tanh$. The largest $n$ is 11000, $\sigma_\varepsilon = 0.3$ and the learning rate is $\eta = 3.6$ for all training processes. We train each neural network until the training losses approach zero. Each experiment repeats 15 times.

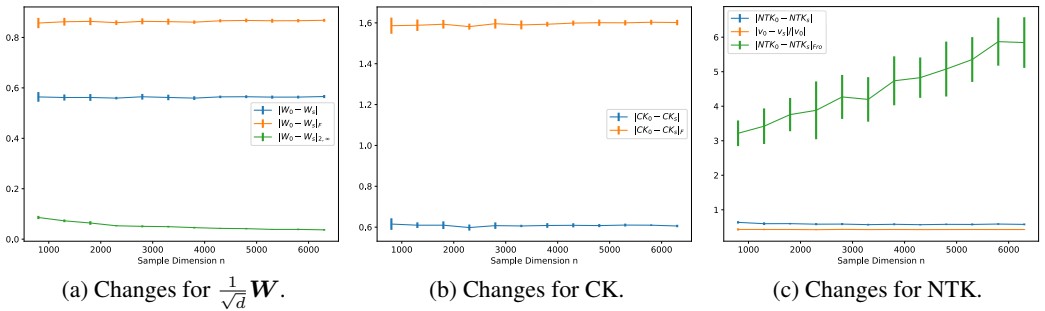

(a) Changes for $\frac{1}{\sqrt{d}}\boldsymbol{W}$.     (b) Changes for CK.     (c) Changes for NTK.

Figure 33: Measuring the change for $\boldsymbol{W}_t$, $\boldsymbol{K}^{\text{CK}}$, the second layer $\boldsymbol{v}_t$ and $\boldsymbol{K}^{\text{NTK}}$ when training NN with SGD for both layers $\boldsymbol{W}_t$ and $\boldsymbol{v}_t$. We fix $d/n = 0.6$ and $h/n = 1.2$ when $n$ is increasing. Here, $\sigma$ is normalized ReLU and the target is normalized $\tanh$. $\sigma_\varepsilon = 0.1$ and the learning rate is $\eta = 1.0$ for all training processes. We train each neural network until the training losses approach zero. Each experiment repeats 12 times.

