# OpenReview forum: "Spectral Evolution and Invariance in Linear-width Neural Networks"
_NeurIPS.cc/2023/Conference — NeurIPS 2023 poster_

### Official Review · Reviewer_zoyL · 2023-06-21

**Soundness:** 2 fair
**Presentation:** 2 fair
**Contribution:** 2 fair
**Rating:** 4
**Confidence:** 4

**Summary:**

This paper studies gradient descent training in single-hidden-layer neural networks in the linear-width regime (i.e., that in which the input dimension, hidden layer width, and number of training datapoints together tend proportionally to infinity). Its primary result is that the bulk spectra of the conjugate and tangent kernels do not change over training in this regime, but at large learning rates an outlier eigenvalue can emerge.

**Strengths:**

The topic of how kernels evolve outside of the lazy regime is timely, and the RMT-inspired setup and results presented should be of interest to the community. The theoretical results appear correct, though I've not checked the proofs line-by-line.

**Weaknesses:**

The manuscript presents a series of interesting observations, but to my taste it sacrifices depth for the sake of breadth. The results on heavy-tailed spectra are thought-provoking, but I think focusing on the emergence of spikes in the spectra would make for a stronger manuscript. If the authors could prove the existence of a BBP-like transition, they would have a strong paper. The current comment on Lines 247-248 comparing to [6] is not very satisfying. At the very least, a more detailed empirical investigation of this phenomenon would result in a more convincing story.

In this vein, the experiments are not very systematic, and as a result cannot clearly disentangle which changes in the setup are required to produce a given phenomenon. For instance, Table 1 contains various settings of batch size and learning rate, but nowhere are these parameters swept jointly over some reasonable range. I have in mind something like the parameters sweeps for learning single-index models performed in recent work by Atanasov et al., "The Onset of Variance-Limited Behavior for Networks in the Lazy and Rich Regimes" (ICLR 2023; note also that this relevant recent work is not cited). The goal here would be to establish under precisely what conditions a spike emerges. As it stands, the authors can make only vague statements.

The transition to using Adam for the heavy-tailed weight experiments is even more drastic: is it possible to observe this phenomenon using a non-adaptive optimizer? I think dissecting this phenomenon is more properly the subject of a separate paper, as it is currently not well-integrated with the other results. The discussion in Lines 286-291 is also rather heuristic, and I don't think the authors provided sufficient evidence for their claim that heavy-tailed spectra naturally emerge in feature-learning networks trained on complex tasks. Moreover, how does "[t]his example explain[] why we can use the heavy tails to discriminate well-trained and poorly-trained large models?" This requires elaboration, and more empirical evidence.

**Questions:**

- In the Introduction, it would be nice to mention recent work on Gaussian equivalence for deep random feature models in the linear-width regime by Schröder et al., "Deterministic equivalent and error universality of deep random features learning" (ICML 2023) and Bosch et al., "Precise Asymptotic Analysis of Deep Random Feature Models" (2023).

- The linear-width regime for deep Bayesian neural networks has recently been the subject of study in the community of researchers using tools from statistical physics to study deep learning, see e.g., Li and Somplinsky, "Statistical mechanics of deep linear neural networks: The backpropagating kernel renormalization" (PRX 2021), Zavatone-Veth et al., "Contrasting random and learned features in deep Bayesian linear regression (PRE 2022), and Cui et al., "Optimal learning of deep random networks of extensive-width" (ICML 2023). I think it would be useful to at least comment upon the Gaussian equivalence results of Cui et al. as a point of reference for how different inference procedures behave in the extensive-width regime.

- In Figure 2 (b-d) and elsewhere, it would be better to label the abscissa with $\eta$ or "learning rate" rather than "lr," for the sake of consistency with the text.

- In Figure 1 and elsewhere, the histograms should be plotted using a log-scaled ordinate. As it stands, the outlier in 1b is nearly invisible.

- Why use the single-index + quadratic model in eq. 8 rather than a (simpler) single-index model?

- The BERT experiments in Figure 5 strike me as mostly decorative, as the setting is not commensurate with the earlier portions of the paper. What insights would the authors argue can be transferred from their MLP experiments to this setting?

**Limitations:**

The paper does not provide adequate discussion either of the relevance of the linear-width setting for practice or of the limitations of their experiments.

---

> ### Author Rebuttal · Authors · 2023-08-10
>
> We thank the reviewer for the thoughtful feedback and additional references on random feature models and deep Bayesian networks under LWR. We will add these references and provide additional comments and comparisons. In the following, we address the comments and questions.
>
> 1. The goal of this paper is to demonstrate empirically that this BBP-like transition we observed appears in neural networks under LWR, both for the weight and kernel matrices. While proving this theorem in particular setups is our ultimate goal, this paper is an empirical study that can help inform our and others' approaches to rigorously proving such a transition. We are familiar with reference [6] where the authors considered a two-stage training process and proved the BBP-like transition for the weight matrix, in a much simpler setting than the one considered here. We will provide a detailed comparison with this result in Section 4.2.
>
> 2. Thanks for your suggestions on improving the readability of our captions and figures. We will change things accordingly. We'd like to point out that our experiments **are** systematic, as follows: we fixed the synthetic dataset, the teacher model, and the two-layer neural networks under the linear width regime; then, we used different optimizers (GD/SGD/Adam) to train the same network. For each optimizer, we tuned the hyperparameters, e.g., learning rate, to observe different spectral behaviors and accuracy after training. For example, Figure 2(b-d) presents how the spikes and eigenvector alignments emerge when we use different learning rates. Although we do not have a theoretical threshold for the learning rate when the spike emerges, we demonstrated these transitions and threshold empirically. In the attachment to the general rebuttal, we show how we tune the hyperparameter (a grid search for learning rate) for GD when presenting Table 1. In the revision, we will add more explanations on how to choose the parameters in the experiments in Table 1 of our paper.
>
> 3. In the heavy-tailed section, our main message is two-fold. In references [55-57], the authors explored a correlation between the heavy tails in the spectrum and the good performance of the NN. They left causality as an open question. In our paper, we rule out general causality: Figure 3 (c-d) shows that there are cases when both the weight and CK matrices appear to exhibit power law in their spectrum, but the neural network does not perform well. Second, we argue that even if not causal, the relationship between heavy-tailed spectra and good performance can be subtle: we demonstrate an example where a neural network with heavy tails generalizes well on the test dataset in Figure 3 (b) and Figure 18. Additionally, we exhibit a case when neural networks benefit from heavy tails: When the teacher model is a multiple-index model with a high intrinsic dimension and the heavy-tailed part is really aligned with the multiple-index directions well, the neural network with heavy tails will generalize well (see Figure 18).  We will move the experiment of Figure 18 to Section 4.3 and clarify our explanations.
>
> 4. The reason we did not choose the log scale is that we want to show that initial spectra are Marchenko–Pastur-type distributions that one could not see in a log scale. We will adopt the reviewer's suggestion, though, and use log-scaled histograms to highlight outliers.
>
> 5. The reason we chose the teacher model as a single-index + quadratic in Eq. (8) is to make it a little more complicated to learn. However, the performances and the spectral properties are similar to the case when the teacher model is just a single-index model. To make the paper more consistent and clearer, we will only present the cases when the teachers are single-index and multiple-index models in our reversion.
>
> 6. The BERT example is important to the point we are trying to make, as other reviewers have noticed. The application of a transfer BERT model onto Twitter140 relies on a mathematical trick allowing us to pretend that this model is a 2-layer feed-forward (FF) network. We can think of any FF network as a composition of two functions, $f(g(x)) $, where $f$ is a classification head and $g$ is the composition of all the hidden layers; in this instance, studying the CK matrix for MLP is equivalent to studying the output of $g$. We can, in principle, do this for any architecture of neural networks. From these experiments, we want to highlight that understanding the spectral evolution of the conjugate kernel matrices for the large language model is crucial for analyzing the pre-trained model and fine-tuning process. First, in our experiments, spikes and heavy tails emerge in the spectrum of conjugate kernel---which is analogous to our toy models with synthetic datasets. Second, Figure 5 indicates that the evolution of the spikes in the conjugate kernel is closely related to the feature learning in the fine-tuning process. In the general rebuttal, we provide additional figures showing how these spikes forget the pre-trained feature and learn new features efficiently, quantitatively corroborating what Chen et al. have qualitatively shown in their preprint [CZZ23].
>
> 7. We will provide an adequate discussion of the advantages and limitations of the linear-width setting. We will especially mention how the linear-width regime has been applied to study the Gaussian equivalence for deep random feature models and deep Bayesian networks. Our linear-width regime is more realistic compared with infinite-width neural networks and it has the theoretical benefit that random matrix theory can be applied in this setting. This linear-width regime is one of the ways to approximate finite but very large neural networks with very large datasets.
>
>  [CZZ23] Chen, L., Zaharia, M. and Zou, J., 2023. How is ChatGPT's behavior changing over time?  arXiv:2307.09009.
>
> We are happy to clarify any concerns or answer any questions that may come up during the discussion period.

---

> > ### Comment · Reviewer_zoyL · 2023-08-11
> >
> > I thank the authors for their detailed response to my concerns and those of the other reviewers. I maintain that the clarity of the paper could be improved by providing more detailed experiments investigating a smaller set of phenomena. This is related to the comment made by Reviewer C57a regarding the need to `zoom in' to the transition region; if the authors can provide a detailed investigation of this phenomenon, it would greatly enhance the paper. Respectfully, I do not think the BERT plots convincingly demonstrate the claim the authors are trying to make. I do appreciate the authors' efforts---I think the changes proposed will improve the paper---so I will raise my score.

---

> > > ### Author Response · Authors · 2023-08-21
> > > **Thanks for the reply**
> > >
> > > Thanks for the reviewer's response and valuable suggestions. Here are some additional responses.
> > >
> > > 1. In this paper, we showed three different spectral evolutions (invariance, emergence of spikes, and heavy tails). We proved the spectral invariance in certain regimes and presented additional experiments on the other two phenomena. From a spectral analysis perspective, we believe it becomes more complete and all these phenomena are indispensable in our paper. Because of paper limitations, we did not provide enough analysis for all three phenomena. But definitely, in our revised appendix, we will provide more experiments on the transition phenomenon for the spikes in the spectra. Following the figures in the attachment of the general rebuttal, we will show a detailed investigation of this transition phenomenon, especially how the learning rate affects the spike's alignment and test error in the transition region.
> > > 2. For the BERT model, we do not want to draw any conclusion but aim to draw the attention of RMT and statistics communities, emphasizing the importance of spectral analysis for even large language models. And we possibly can use spectral analysis to understand how the model learns features. Also, like Figure 4, we want to show that our LWR can be considered in more practical models. It is possible to apply RMT to analyze these large models for future work.

---

### Official Review · Reviewer_C57a · 2023-07-06

**Soundness:** 3 good
**Presentation:** 3 good
**Contribution:** 3 good
**Rating:** 7
**Confidence:** 2

**Summary:**

This paper examines the spectrum of the conjugate kernel and neural tangent kernel of real networks before and after training. Based on analyzing the spectrum in some experiments, several observations about how the spectum evolves during training are made. In particular, there seems to be different phases depending on the optimizer used and learning rate which effect how the spectrum changes from initialization. Understanding this spectrum-evolution phenomenon may lead to a better understanding of how neural networks train, and shed light on important questions like understanding differences between ADAM vs GD vs SGD.

**Strengths:**

* The review of existing literature is really well done (there is even a more detailed review in the appendices). This puts the current work in context quite nicely and makes this paper a very accessible place to start learning about these spectral questions.
* The experimental plots are nicely done and clearly showcase the behaviour being discussed (the Q-Q plots also help a lot with the understanding). There are also numerous other experiments in the appendix. This again makes this paper a great resource for these kinds of experiments.
* I am not an expert on this area, but to me the way the different phases of spectral evolution are laid out (i.e. what they call "Invariant Bluk" vs "Bulk+Spike" vs "Heavy tail") is new to me. (I have heard of this outlier thing happening before for the Hessian...I am not sure how related that is). To me this paper looks like the first step towards building a phase diagram of how choices about optimizer/trianing effect the spectrum which would be a really useful thing to have.
* I found the conclusion about learning rate (e.g. Figure 2) the most impressive conclusion here, and to me this may be the most actionable of all the various phenomenon observed here. This is a nice conclusion to draw and could pave the way for more results in this direction.

**Weaknesses:**

* One might complain that the theory developed here is a bit weak, in that the theoretical results are all inequalities which are (probably) not sharp. However, since the paper is mainly empirical I actually think these theoretical results are reasonable here.
* I am not an expert on this area, so I was not able to tell how much of the results here were expected/known already. Hopefully at least one other reviewer is able to give that context.
* It is not clear how robust the experimental results are: in the given plots the phenomenon seem to be happening as described, but maybe there could be some extra discussion about "edge cases" between the various phases and discussion about when things break down (e.g. when N is too small, or there are not enough examples etc)

**Questions:**

* Is there any possibility to describe how the outlier eigenvalues are evolving in time during training? Particularly when there is only one outlier eigenvalue, this would be an interesing bit of theory to try and understand.
* Is there any relationship between the spectrum of the CK/NTK and the spectrum of the Hessian?

**Limitations:**

* One limitation that always comes up in these limiting type things is the question of how large real neural networks have to be for the theory to actually work. Some discussion or experiments specifically addressing this (e.g. showing the error in the predictions as a function of network size) could help explicitly address this.

---

> ### Author Rebuttal · Authors · 2023-08-10
>
> We thank the reviewer for the positive evaluation and thoughtful feedback. In the following, we address the technical comments and questions.
>
> 1. Our experiments are robust to dimensions as long as the width $h$, sample size $n$, and feature dimension $d$ are large. Anecdotally, when these dimensions are around several hundred is enough. As in random matrix theory, when the aspect ratios are fixed and the dimensions are large enough, the eigenvalue distribution is close to the limiting law, and it is very stable. We will present additional experiments for different aspect ratios to show that our phenomena are stable for different aspect ratios when dimensions are large enough.
>
> 2. We thank the reviewer for asking about the evolution of the outlier through training. We have not checked this evolution in the paper since we only present the emergence of the outliers after training. In the attachment of the global rebuttal, we present the evolution of the outlier during the training process when there is only one outlier eigenvalue. There are some interesting phenomena in this evolution. We observe the training error first increases and then decreases. We believe this regime corresponds to the catapult phenomenon in reference [47].  And the outlier for the kernel matrix first progressively increases and then oscillates around some value before convergence. We agree this will be an interesting theoretical direction to further explore.
>
> 3. We thank the reviewer for asking the question related to Hessian. In fact, NTK has a close relationship with the Hessian with square loss function. There are two parts of the Hessian and one of these parts is the Fisher information matrix which has the same non-zero eigenvalues as the NTK matrix. This is another motivation for analyzing the spectrum of the NTK matrix through training. We can have a better understanding of the spectrum of the Hessian matrix and the loss landscape, e.g., the relationship between sharpness and generalization, during different training processes.
>
> 4. We agree that our theory requires asymptotic limits. However, as classical results in random matrix theory, the eigenvalue distributions of random matrices will be very close to the limiting law even when the dimension is several hundred. We believe that there will be some **non-asymptotic** results for our empirical observations, i.e. our empirical results should hold practical neural networks whose widths and sample size are large but finite. Additional discussion and experiments (as dimensions are gradually growing) will be added to explain this problem.
>
> We would be happy to clarify any concerns or answer any questions that may come up during the discussion period.

---

### Official Review · Reviewer_DCEt · 2023-07-10

**Soundness:** 3 good
**Presentation:** 3 good
**Contribution:** 3 good
**Rating:** 7
**Confidence:** 3

**Summary:**

The paper carries out an empirical study of the spectral properties of finite-width neural network kernels after training, and compares them to their infinite-width counterparts to assess if there is an improvement. They study these changes with respect to various hyperparameters such as learning rate and optimizer. The results suggest that for large learning rates, where the linear dynamics is not a valid approximation, they show that feature learning emerges by inspecting the distribution of the final spectra. They identify two different spectral property that distinguish feature learning from lazy regime: emergence of low-rank spikes in the spectrum and heavy-tailness. They explore when these properties emerge and provide theoretical arguments for these observations.

**Strengths:**

The paper is well-written and easy to follow. The experiments are complimented well with theoretical arguments, and the discussion is sound. The Supplementary Material also includes nicely organized discussion of previous works and additional experiments and discussion.

In summary, they introduce the well-known fact that the small-learning rates make a NN with NTK parameterization stuck around its initialization and derive theoretical results to show that the bulk of the spectrum remain invariant under lazy training. In my opinion, the main contribution is that they find two different ways feature learning can happen: emergence of low-rank outliers in the spectrum with large learning rates with GD optimizers and heavy-tailed spectrum with adaptive optimizers.

Furthermore, for GD/SGD, they make a theoretical connection for how low-rank outliers emerge with a phase transition in RMT (ref. [9] in main paper.)

Finally, the relation of heavy-tailed spectrum to generalization has been discussed.

**Weaknesses:**

1. While the paper brings valuable insights, it is highly restricted by the 2-layer neural networks and synthetic dataset with a particular target. It would be useful to perform experiments on simple real datasets such as MNIST to show that spikes related to number of classes also emerge in the spectrum. For example, in neural collapse phenomenon, one can see that weight matrices of last layer acquire low rank structure that can be tracked to the number of classes.

2. Such experiments can also shed light on how full-batch vs. stochastic GD differ in terms of the after-training spectrum. Currently, I do not think there are any insights related to how SGD helps feature learning since GD experiment does not have a large LR analogue.

3. Is there an intuitive explanation how adaptive optimizers cause heavy-tailed spectrum in CK? More discussion around this phenomenon could be very helpful.

**Questions:**

1. It is hard to parse Figure 1, first panel shows weight spectrum for GD with LR 5, second panel shows CK spectrum for SGD with LR 22 and last one shows CK spectrum for ADAM. Something like CK spectrum for Case 2, 3, 4 would be more helpful.

2. In Figure 4, is there a reason why GD does not show the same properties as SGD? With large enough LR, could GD also show the same heavy-tailness? Also, it is not clear how adaptive optimizer qualitatively differs from SGD.

**Limitations:**

A limitations section is missing.

---

> ### Author Rebuttal · Authors · 2023-08-10
>
> We thank the reviewer for the positive evaluation and thoughtful feedback. In the following, we address the technical comments and questions.
>
> 1. **Q: Real datasets such as MNIST?** We will present experiments of the MNIST dataset and show that the spikes in the kernel matrix are related to the different classes of the dataset. In the attachment of our global rebuttal, we present the top principal components of pre-trained transformers before and after fine-tuning. Similarly, with the neural collapse phenomenon, we can see that these principal components correlate to the different classes after fine-tuning training. This should be consistent with the MNIST experiments the reviewer describes in the question.
>
> 2. We thank the reviewer's suggestion for additional experiments to compare full-batch GD and SGD. In the attachment of the global rebuttal, we present additional simulations for GD training with different learning rates and the maximal learning rate we can use in GD training. Especially for GD training with very large learning rates, we can observe the emergence of an outlier in the spectra after training, but the performance is not as good as SGD training with large learning rates. And in this case, the training dynamic with GD is unstable.
>
>    Besides, we also run a grid search for the learning rate when training with GD/SGD, and we do not figure the same heavy-tailed spectra as adaptive methods showed. We can observe the emergence of heavy tails if we apply sufficiently large learning rates for GD at the early stage of training, but we must adjust the learning rate later to ensure the convergence of the training loss. This is the heuristic reason we need adaptive optimizers to obtain heavy-tailed spectra in weight and CK matrices after training.
>
> 3. So far, we do not have a sufficient understanding of what causes a heavy-tailed spectrum after training. But we believe that the heavy-tailed spectrum in CK is due to the heavy-tailed spectrum of the weight matrix. And its heavy-tailed spectrum indicates that the weight matrix is moving far away from the initialization which possibly leads to good feature learning. Moreover, a large learning rate at initialization is important to feature learning, and adaptive optimizers will then guarantee convergence to a global minimum. Our experiments in Figure 18 provide more insights into the adaptive optimizers, heavy-tailed distribution, and how this heavy-tailed distribution related to the multiple-index teacher model. We will move this part to the main part for more discussion.
>
> 4. In Figure 1, we only presented part of the spectral results for different cases. In Appendix B.2, we additionally present all the spectra of the weight matrices, CK, and NTK matrices before and after training in Case 1-4 separately. We will mention this in the caption of Figure
>
> 5. We will present a section about limitations in the conclusion part of our paper for completeness.
>
> We would be happy to clarify any concerns or answer any questions that may come up during the discussion period.

---

> > ### Comment · Reviewer_DCEt · 2023-08-21
> >
> > I thank the authors for their detailed rebuttal and clarifications. I will raise my score accordingly.

---

### Official Review · Reviewer_m9fj · 2023-07-11

**Soundness:** 3 good
**Presentation:** 4 excellent
**Contribution:** 3 good
**Rating:** 5
**Confidence:** 3

**Summary:**

This paper analyzes the spectral properties of feedforward neural networks under a student-teacher setting. A linear-width regime is considered, where the sample size and network width grow comparably with the input feature dimension. The paper shows that the spectra of weight and kernel matrices are invariant under training, and that outliers occur in large step size regimes. Heavy-tailed spectra are also discussed with their relation to generalization. Evidence from both synthetic and real-world datasets is shown.

**Strengths:**

- Proposed LWR as a novel perspective for analyzing feedforward networks, and presented an interesting analysis of NN spectral invariance, potentially providing insights into NN feature learning.
- Evidence from both synthetic and real-world datasets enhance the argument.
- The presentation of the paper is clear.

**Weaknesses:**

- The proposed LWR feels under-explained. Please see "Questions". Otherwise I think this is a solid papar.

**Questions:**

- In the proposed LWR, how should one understand that the input feature dimension $d$ goes to infinity? This feels somewhat counterintuitive, compared with e.g. the infinite-width regime.
- To what extent does the observed spectral invariance depend on the LWR? I would assume this does not occur under the infinite-width regime.

**Limitations:**

n/a, theoretical work

---

> ### Author Rebuttal · Authors · 2023-08-10
>
> We thank the reviewer for the positive evaluation and thoughtful feedback. In the following, we address some of the technical comments and questions.
>
> 1. **Q: Why feature dimension goes to infinity?** The LWR, where the input feature dimension $d$ goes to infinity proportionally to the sample size $n$, is a classical setting in high-dimensional statistics, and provides important insights for real-world datasets. This is in contrast to the infinite-width regime, in which we are already in the asymptotic limit for width at first.
>     We believe the LWR is a better approximation of real-world datasets and practical neural networks compared with the infinite-width regime since the dimension $d$ in real datasets is very large and we should allow the dimension of the feature space to grow with the sample size. In real networks, $d$ is not infinite but can depend on $n$: with a larger $n$ we may be able to use higher-dimensional features. For instance, the feature dimension of the ImageNet dataset is usually cropped to $d=224\times 224$ for learning.
>
> 2. **Q: How depends on the LWR?** Our theory of spectral invariance depends on the LWR, but it can be extended to the infinite-width regime easily. In fact, in the infinite-width regime, we can have sharper estimates of the changes of the weight matrix and each neuron as long as the width $h$ is much larger than sample size $n$. The main difficulty of our theory is to extend the result of the spectral invariance to LWR which is a more realistic regime for neural networks. Besides, although we can show that each neuron is close to the initialization under the infinite-width regime, we cannot get a limiting eigenvalue distribution of the weight matrix when $d$ is fixed and $h\to\infty$. Only when we consider the LWR, we can claim that the limiting eigenvalue distribution of the weight matrix is an invariant Marchenko–Pastur distribution through GD training with a small learning rate.
>
> We would be happy to clarify any concerns or answer any questions that may come up during the discussion period.

---

### Official Review · Reviewer_jctx · 2023-07-19

**Soundness:** 3 good
**Presentation:** 2 fair
**Contribution:** 3 good
**Rating:** 6
**Confidence:** 3

**Summary:**

Considerable attention in deep learning theory has been given to the analysis of neural networks in the kernel regime, which neural networks approach as they approach infinite width. While this analysis has led to insights, there is an important gap between the theorized generalization performance of neural networks in this infinite width regime and the practical performance of finite-width neural networks, which typically perform better.

This paper studies simple, two-layer neural networks in the more realistic *linear-width regime* (LWR), in which sample size, input feature dimension, and the width of the layer approach infinity at comparable rates. In several analyses, the authors target the following question: *How do the spectra of the NN's weight and kernel matrices evolve during the training process?* The authors compare to the kernel regime as a benchmark and analyze changes in spectra that occur through training with different gradient descent algorithms .

For their first simple experiment (Section 3), they find that the spectra of the weight, conjugate kernel (CK) and neural tangent kernel (NTK) matrices remain invariant during training for gradient descent (GD) and stochastic gradient descent (SGD) with a low learning rate. This is consistent with the Lazy Training (LT), and indicates that the network is not outperforming a kernel machine. However, for SGD with a large learning rate, a spike in the spectral distribution emerges after training, which indicates improvement over lazy training. They note that this spike is consistent with previously published results that demonstrate that learned features may be indicated by an outlier in the spectra with a large singular value.

They classify the spectral distributions into three categories: invariant bulk, spikes outside the bulk, and heavy-tailed distributions. They prove that the invariant bulk phenomenon is expected under certain assumptions. They then empirically analyze the spike phenomenon, and demonstrate that as the learning rate increases, there is a threshold at which the spike emerges, which corresponds with greater alignment and suggests feature learning. They then provide an explanation for the relationship between heavy tailed spectra and better generalization performance. Lastly, they perform some analyses on CNNs trained on more a more natural dataset, CIFAR-2, as well as BERT with fine-tuning on Sentiment140.

**Strengths:**

The authors tackle an important question, which is to understand how the structure of neural networks change through the training process. To get a handle on it, they analyze the spectra of the weight and kernel matrices of simple neural networks on controlled datasets. They find some interesting phenomena that relate the spectral properties to learning rate and generalization performance.

**Weaknesses:**

The paper would benefit from more effort put towards making clear the meaning and significance of the experiments and results. Why look at the spectra? What is the significance of the three classes of spectral distributions that are observed? (The answer to the latter question is in the paper, but it is somewhat buried. This should be put front and center and made very clear).

The figures and captions in the paper could be clearer. Oftentimes, variables are used as labels. The reader forgets which variables correspond to what. It would be better to use English names, or more descriptive properties.

The specific primary findings should be stated more clearly. The statements provided in the introduction are general, and don't provide much insight into the conclusions. It takes work for the reader to tease out these conclusions.

**Questions:**

- What can we understand about neural networks, generally speaking, by looking at the spectra of their weight matrices and kernels? Why use this approach, rather than analyzing other properties of the weights and network structure?
- How do the results found here advance our theoretical understanding of how deep networks learn?
- How can these results be used in practice?


**Limitations:**

The authors note the limitation of the FWR. They do not state limitations of the spectral approach to analyzing networks.

---

> ### Author Rebuttal · Authors · 2023-08-10
>
> We thank the reviewer for the positive evaluation and thoughtful feedback. Per the reviewer's suggestion, we will provide more detailed captions for the figures to help readers. We will also clearly state our primary findings and conclusions in the introduction. We have summarized this work's main contributions and insights in the global rebuttal for clarification. In the following, we address the technical comments and questions.
>
> 1. **Q: Why look at the spectra? Theoretical understanding?** The spectral properties of the neural networks are a key component of understanding the explainability and dynamics of feature learning in deep neural networks. We believe that a spectral analysis of weight and kernel matrices can be seen as one of the first, fundamental steps in understanding the performance of neural networks.
>
>       There are many previous works that use the spectral properties of either the weight or kernel matrices of neural networks (there are many references [28, 55-57,60,70,71,82]). Intuitively, we can think of the eigendecomposition of the CK and NTK as describing the variance of the underlying feature space, akin to PCA. The emergence of an outlying eigenvalue tells us that the total explained variance of that feature space is becoming concentrated along one direction. In our alignment experiments (see Figures 1(b) and 2(a)), we show that this direction corresponds to the target space of the neural network. This alignment could be thought of as an operant definition of feature learning itself.
>
>      While there are alternative approaches to the spectra, this analysis is simple and theoretically motivated. Furthermore, in Figure 2(b-d), the emergence of the outlier and strong alignment are highly related to the choice of learning rates, which indicates that the spectral properties we studied here help us understand how we choose the correct optimizer and hyperparameters for efficient training. In conclusion, our empirical results of spectral evolution provide a promising way for future theoretical understanding of how deep networks learn features during training.
>
> 2. **Q: Applications in practice?** Based on our empirical results, we emphasize that understanding the spectral properties of the weight and kernel matrices is essential to understanding feature learning. For example, in Figure 5, we computed the top two principal components of the CK matrix in the BERT model before and after fine-tuning, and we can see how the model forgets the pre-trained features and learn new features in the fine-tuning process. In addition, our results indicate that choosing optimizers and hyperparameters affects the spectral behavior and generalization of neural networks. Finally, [55-57] found a correlation between the occurrence of heavy tail spectra and generalization. Our experiments rule out that this relationship is causal. Our example in Figure 18 empirically shows that we need to analyze additional spectral properties, e.g., how well features align with the eigenspace in the heavy-tailed part, to test if the neural network generalizes well or not.
>
> We are happy to clarify any concerns or answer any questions that may come up during the discussion period.

---

> > ### Comment · Reviewer_jctx · 2023-08-11
> >
> > I thank the authors for the thorough rebuttal. My primary concerns with the paper regard clarity. Other reviewers have noted similar challenges with parsing the plots and remarked that the paper "sacrifices depth for the sake of breadth." I believe the results are important and that the work is sound. I agree with reviewer zoyL that, for my taste, I would prefer to see a more focused set of experiments on a smaller set of phenomena. Nonetheless, given the scope that the authors have chosen to present in this paper, I believe the changes proposed in the rebuttal will improve the paper, and I have increased my score accordingly.

---

> > > ### Author Response · Authors · 2023-08-21
> > > **Thanks for your feedback**
> > >
> > > Thanks for your thoughtful response and valuable suggestions. We will balance the depth and breadth in our final version of the paper. In our revision, we will provide more experiments on the transition phenomenon for the spikes in the spectra of kernel and weight matrices. Following the figures in the attachment of the general rebuttal, we will show a detailed investigation of the phase transition phenomenon for spikes, especially how the learning rate affects the spike's alignment and test error in the transition region. This will provide a more focused case study of experiments on our second phenomenon.

---

### Author Rebuttal · Authors · 2023-08-10

**Overall Summary:**

We would like to thank all reviewers for their evaluation of our work and their helpful comments. The key contribution of our work is to describe how the spectral evolution of both weight and kernel matrices changes during different training procedures. We focus on a simple two-layer neural network under a linear-width regime (LWR). These empirical results open a promising way to understand the training dynamic and feature learning of neural networks via random matrix theory. We believe our results will be impactful for both the random matrix theory and optimization communities by providing experimental evidence that can inform future theoretical studies. We briefly summarize our main results:

1. We observe the invariant spectra of weight and CK matrices through training with GD/SGD and small learning rates, which indicates the performance of this kind of training is still close to the kernel regime. We theoretically justify this observation for invariant weight and kernel matrices under certain assumptions by using the global convergence of GD under LWR.

2. We observe a strong alignment between the eigenvector corresponding to the outlying largest eigenvalue and the teacher model when training with a large learning rate. We demonstrate how to find the threshold for a large enough learning rate by showing a BBP-like transition in the spectra of the weight and kernel matrices when increasing the learning rate. The importance of this BBP-like transition is critical, as it is known to be indicative of feature learning, as explained in Response 1 to **Reviewer jctx**. This will also help us understand how to choose the learning rate to attain feature learning.

3. Our experiments rule out a causal relationship between the occurrence of a heavy-tailed spectrum for the weight matrices and a good generalization.  This complements the work of Mahoney et. al. [55-57] in reference where the authors had observed a strong correlation between the two; our work can be considered a limitation of the phenomena underlying that trend, while at the same time, through the example in Figure 18, we confirm the existence of a relationship and heuristically explain why this neural network benefits from heavy tails.

4. We investigate the properties of weight and kernel matrices of larger models to demonstrate the phenomena observed in our toy examples. Therefore, our spectral analysis has the potential for high impact as we can investigate feature learning and training dynamics for different optimization algorithms using models used in applications.

5. Given **Reviewer zoyL**'s comments, we want to emphasize the fact that our BERT experiment is motivated by a connection to the spectral analysis we do on the feed-forward neural networks. Please see item 6. in our response to **Reviewer zoyL**. To reinforce our point here, we are including two PCA plots in Figure 3 of the attachment which are part of a related project we are working on. Due to authorship restrictions, these plots cannot appear in our revision, but we hope they can help inform our responses regarding spectral theory and feature learning.

**Additional Experiments in the Attachment:**
1. Following the experiments in Figure 2 in our paper, we present the training dynamics of SGD training with a learning rate larger than the BBP-like threshold in Figure 2. We present the evolution of the largest eigenvalues (outliers) of CK and NTK matrices respectively during this training process. There are two phases of this evolution: at the early stage, the largest eigenvalues progressively grow, and then these outliers start oscillating before convergence.
2. Following the experiments in Table 1 of our paper, we further present the GD training with a grid search for all possible learning rates which ensures global convergence after training. We did not observe heavy tails for all these learning rates. However, analogously to the SGD case in Figure 2, the emergence of an outlier also appears when gradually increasing the learning rates. We then separately present the spectral behaviors of weight matrices when using a small learning rate and the maximal learning rate for GD training.
3. Following the BERT experiment in our paper, we further present the alignment between the top two principal components of the CK matrix of a large language model before and after fine-tuning.


We would be happy to clarify any concerns or answer any questions that may come up during the discussion period.

---

### Decision · Program_Chairs · 2023-09-21

**Decision:**

Accept (poster)

**Comment:**

This paper analyzes the NTK spectra of single-hidden-layer neural networks in the regime in which the number of samples is proportional to the width. A number of interesting empirical observations are obtained, and among other things it is argued that the bulk spectrum remains invariant but a spike can emerge for large learning rates.

The reviewers generally found the topic relevant and the findings to be interesting to the community. The breadth of the investigation is probably the paper's main strength and weakness. The number of phenomena identified and explored paints a rich landscape worthy of significant further study, with many interesting observations already identified here; on the other hand, no single topic was explored in sufficient depth to tell a complete story about it. This tension led to some mixed perspectives in the reviews.

There is no directive that a NeurIPS paper must present a deep dive into a very narrow topic, and the question is whether the broader approach undertaken here yields a paper that has non-trivial and novel insights that will interest the community and inspire future work. It seems the answer to that question is almost certainly yes and I therefore recommend acceptance.